# Behavioral and neuro-cognitive bases for emergence of norms and socially shared realities via dynamic interaction

Kiri Kuroda [1,2,3,9,11], Yukiko Ogura[4,11], Akitoshi Ogawa [5,10], Tomoya Tamei[6], Kazushi Ikeda [7] & Tatsuya Kameda [5,8,9 ✉]

In the digital era, new socially shared realities and norms emerge rapidly, whether they are beneficial or harmful to our societies. Although these are emerging properties from dynamic interaction, most research has centered on static situations where isolated individuals face extant norms. We investigated how perceptual norms emerge endogenously as shared realities through interaction, using behavioral and fMRI experiments coupled with computational modeling. Social interactions fostered convergence of perceptual responses among people, not only overtly but also at the covert psychophysical level that generates overt responses. Reciprocity played a critical role in increasing the stability (reliability) of the psychophysical function within each individual, modulated by neural activity in the mentalizing network during interaction. These results imply that bilateral influence promotes mutual cognitive anchoring of individual views, producing shared generative models at the collective level that enable endogenous agreement on totally new targets–one of the key functions of social norms.

[1] Japan Society for the Promotion of Science, Chiyoda-ku, Tokyo 102-0083, Japan. [2] Center for Adaptive Rationality, Max Planck Institute for Human Development, 14195 Berlin, Germany. [3] Institute for Research in Business and Economics, Faculty of Economics, Meiji Gakuin University, Minato-ku, Tokyo 108-8636, Japan. [4] Department of Mechano-Informatics, The University of Tokyo, Bunkyo-ku, Tokyo 113-0033, Japan. [5] Brain Science Institute, Tamagawa University, Machida, Tokyo 194-8610, Japan. [6] Department of Robotics, Ritsumeikan University, Kusatsu, Shiga 525-8577, Japan. [7] Division of Information Science, Nara Institute of Science and Technology, Ikoma, Nara 630-0192, Japan. [8] Center for Experimental Research in Social Sciences, Hokkaido University, Sapporo, Hokkaido 060-0810, Japan. [9] Present address: Department of Social Psychology, The University of Tokyo, Bunkyo-ku, Tokyo 113-0033, Japan. [10] Present address: Faculty of Medicine, Juntendo University, Bunkyo-ku, Tokyo 113-8421, Japan. [11] These authors contributed equally: Kiri Kuroda, Yukiko Ogura. ✉email: tkameda@l.u-tokyo.ac.jp

In modern societies where social media connect a huge number of people in near-real time, shared realities[1,2] and norms[3–6] develop rapidly within and across groups. Examples include the formation of collective prejudice against foreigners during the COVID-19 pandemic[7] and the broadening consensus for decarbonized transport and energy[8], not to mention the moral and political divides between social sectors[9]. These socially shared realities affect what people see or attend to in the focal situation, how and to whom what they've seen is communicated, and when and how people edit their own specific behaviors[10–13]. Accordingly, these realities fundamentally affect both the processes and outcomes of collective decision-making at various levels, ranging from pairs to groups to organizations to societies[14,15]. Whether such shared realities are beneficial or harmful, it is noteworthy that they are not mere duplications of exogenous extant norms but emerge endogenously from people's interactions, and thus they may be hard to change once established[3,6,16–18]. Most previous studies have addressed how people conform to (or deviate from) fixed, extant norms (see[19–22] for review); our understanding of the dynamic emergence of norms through interaction, therefore, remains elusive. Using neuro-cognitive techniques, we aim to shed light on the computational mechanisms underlying the formation of norms as shared realities.

The term social norm refers to a standard or pattern of behavior that is accepted in or expected of a group[6,18,23], constituting a shared reality about what is normal in a given situation. Cialdini and colleagues[24] provided a useful distinction about this concept: Descriptive norms characterize the perception of what most people do (norms about "is"), and injunctive norms characterize the perception of what most people approve or disapprove of (norms about "ought"). Although these notions are often conflated, here we focus on a descriptive norm, which ordinarily sustains an injunctive norm in the real world[25].

To set the stage for our argument, we first outline two representative experimental paradigms to study descriptive norms. The formation of descriptive norms has often been studied following Solomon Asch's conformity paradigm[26]. Using an objectively verifiable perceptual task, Asch examined how participants were influenced by unresponsive majorities (confederates) that repeatedly endorsed incorrect answers. The participants conformed to the majority view publicly, but once the group was dissolved, they reverted to their private (correct) answers. Where this unilateral-influence paradigm has been used in neuroscience research about social decision making[21,22], it has been assumed that social standards are exogenous and fixed[27,28].

In another classic study, Muzafer Sherif investigated norm formation through bilateral influence[18]. Participants collectively viewed a spot of light in a dark room and reported how much the spot moved. When they heard each other's answers, their judgments converged and remained so even when they performed the task later individually. Sherif argued that the participants developed and internalized perceptual norms (i.e., common frames of reference about what is seen and how it is described)[15,29]. Sherif's bilateral-influence paradigm is insightful but silent about the computational mechanisms underpinning the emergence of perceptual norms[30]. Because it employed an optical illusion wherein a static spot of light is perceived to be moving, it is impossible to define a psychophysical function (a psychometric relationship between objective features of a stimulus and the perceptions/judgments about the stimulus[31]) that underpins each individual's responses.

Whereas the Asch and the Sherif paradigms can highlight the difference between public compliance and private acceptance and/or between normative and informational social influence (e.g., whether the behavioral convergence persists even after interaction)[25,32], here, we aim to address a finer distinction—whether the interaction affects only the convergence of people's overt behaviors or also leads to sharing of the covert psychophysical functions that generate their behaviors. This distinction is critical, because only the latter shared generative model enables endogenous agreement on new targets beyond the initial learning set. We believe that such a generative nature[33] is a fundamental characteristic of social norms.

This prompts important questions: how do people develop such shared realities through interaction, whereby norms function as generative rules to regulate overt behaviors? What type of interaction is needed to foster these generative norms, and how persistently and reliably do they modulate individual behavior after interaction?

To address these questions, we investigated the behavioral and neurocognitive mechanisms that underpin the formation of perceptual norms through dyadic interaction, using a dot-estimation task. First, in a laboratory behavioral experiment, we had pairs of participants perform the task and tested these hypotheses: (H1) Dyadic interaction causes convergence of not only participants' overt behaviors but also their covert psychophysical functions, within a pair; (H2) Dyadic interaction stabilizes the psychophysical function within each individual after interaction.

H1 addresses the critical distinction between the overt and covert levels of convergence in formation of perceptual norms as shared realities. H2 addresses the stabilization of the covert psychophysical function through interaction. We believe that an increase in stability (reliability) of the generative function within each individual can have important consequences for accuracy of choices, though this point has rarely been studied. To illustrate, take the example of a weight scale with low validity but high reliability of measurement, which always yields a value 10% less than the actual weight. This scale is inaccurate numerically but still useful for properly ordering new as well as old items in terms of weight. If the stability of a psychophysical function increases through interaction, this could serve the individual by increasing the accuracy of ordering in perceptual judgments and consequently would improve decisions that depend on order by an individual as well as by groups of individuals[15,34].

We conjectured that reciprocity would be a key to facilitating covert-level convergence. Using a perceptual task, Mahmoodi et al. (2018)[35] showed that people were more susceptible to the judgment of a partner who was susceptible to (i.e., responded to) their own judgment during interaction. Reciprocity also operates in various social situations (e.g., self-disclosure[36], interpersonal attraction[37], negotiation[38], cooperation[39]) and is regarded as a core feature of human sociality[40]. Yet, beyond matching of overt behaviors, it remains unknown whether and if so how reciprocity contributes to the emergence of shared realities at the level of covert psychophysical function.

Thus, whereas the pairs estimated freely in the first (laboratory) experiment, we intervened in the interaction mechanics in functional magnetic resonance imaging (fMRI) and online behavioral experiments. We had participants interact with two partners (computer agents), whose behavioral patterns were programmed to follow either Asch-type (unresponsive/one-sided) or Sherif-type (responsive/reciprocal) interactions and tested the behavioral hypothesis using a time-series model: (H3) Reciprocal concession during interaction (as implemented with the Sherif-type but not with the Asch-type interactions) stabilizes the covert psychophysical function within each individual after interaction.

In the fMRI experiment, we addressed how reciprocity affects the neural activity that modulates formation of shared realities. Past fMRI studies have shown that cognitive perspective taking plays a key role in shaping interaction. For instance, when interacting with others in a strategic game[41,42] or in group decision making where coordination or trust was needed[43,44], participants developed internal models about how others decide using the mentalizing network[45], which involves the right

temporoparietal junction (RTPJ) and the dorsomedial pre-frontal cortex (DMPFC). These regions are associated with the ability to infer others' agency and mental states[42,46–56], and also to make decisions on behalf of others[42,57–59]. Here, we conjecture that participants adjust their estimations during interaction by anchoring their views on the partner's views via perspective taking. We reason that a reciprocating (Sherif-type) partner would be more likely to be referred to mentally, that is, to be the target of the participant's perspective taking during interaction, which may help stabilize each participant's covert psychophysical function. We thus derive our fourth hypothesis: (H4) Activity of the mentalizing network during bilateral influence modulates stabilization of the covert psychophysical function within each individual after interaction.

The results of the laboratory behavioral, fMRI, and follow-up online behavioral experiments support all four hypotheses.

## Results

**Laboratory behavioral experiment**. Participants observed random dots for 0.8 s and estimated the number of dots (Fig. 1a). No feedback was given to participants about their accuracy.

In Phases 1 and 3 (hereafter solo phases), participants performed the task individually. In Phase 2, participants were randomly assigned to an individual or pair condition. Participants in the individual condition performed the task individually again. In the pair condition, two participants observed the same dots and answered independently, and they then shared their estimates. Pairs were instructed that they did not need to give similar estimates and that rewards would depend on individual estimation accuracy. Participants remained anonymous to each other and were allowed no verbal communication.

**Modeling a psychophysical function**. We approximated participant $i$'s perceptual response (estimated number of dots that reflect perceptual experience per se and response bias) at trial $t$, $Est_i(t)$, using a linear model:

$$Est_i(t) = w_i \times DotNum(t) + \varepsilon, \quad (1)$$

where $w_i$ denotes the participant's estimation weight, $DotNum$ denotes the number of dots, and $\varepsilon$ denotes Gaussian noise (Supplementary Fig. 1; see Supplementary Table 1 for performance of this model compared to a log-linear model defined in Eq. 3). The estimation weight is less (greater) than 1 if the participant underestimates (overestimates) the number of dots in his/her perceptual response. We calculated the estimation weights of the paired participants in the two solo phases using the maximum likelihood method. Figure 1b shows that participants exhibited an underestimation bias[60] in Phase 1, [$M = 0.915$: one-sample $t$ test: $t_{(41)} = 2.75$, $P = 0.009$, Cohen's $d = 0.42$] and retained this bias uncorrected in Phase 3 after the interaction [$M = 0.912$: one-sample $t$ test: $t_{(41)} = 3.08$, $P = 0.004$, Cohen's $d = 0.48$].

**The covert psychophysical functions converge within pairs through interaction (H1)**. For each pair, we calculated the absolute difference in their estimation weights for the two solo phases. The mean difference decreased significantly after the interaction in Phase 2 [Fig. 1c left; paired $t_{(20)} = 2.81$, $P = 0.011$, Cohen's $d = 0.61$].

However, mere task repetition with another participant rather than actual interaction could have yielded this convergence. We created shuffled pairs who were not partnered with one another and compared each shuffled pair's difference in estimation weights between Phases 1 and 3. Although a significant difference was found between the phases owing to the large sample size [Fig. 1c right; paired $t$ test: $t_{(839)} = 3.03$, $P = 0.003$, Cohen's $d = 0.10$], the decrease was larger in the real pairs than in the

shuffled pairs [Welch's $t$ test: $t_{(21.2)} = 2.27$, $P = 0.034$, Cohen's $d = 0.49$]. We also conducted the same analysis by creating pairs from participants in the individual condition and confirmed that the results remained unchanged (Supplementary Fig. 2). These results support H1's claim that interaction causes convergence of the covert psychophysical functions within pairs.

**The covert psychophysical function stabilizes within each individual through interaction (H2)**. The above results indicate that interaction yielded convergence at the pair level (Fig. 1c) but did not correct underestimation bias at the individual level (Fig. 1b). However, even if estimations remain biased numerically, interaction may stabilize the psychophysical function within each individual. As in the scale metaphor, such stabilization could improve accuracy in ordering targets.

To index the stabilization of participants' estimation weights, we analyzed temporal changes in estimation weights using state-space modeling[61]. This model assesses how estimation weights change over time, while considering autocorrelation between trials and observational random noise (see Fig. 1d for illustration). We assumed that the estimation weight at trial $t$, $w(t)$, was sampled with observational noise from a latent state, $\mu(t)$. The latent state $\mu(t)$ was assumed to be sampled with system noise ($\sigma$) from $\mu(t-1)$ in the preceding trial (see Methods section and Eqs. 4–7 for details). A smaller $\sigma$ indicates that the participant's estimation weight is more similar to that in the previous trial. We thus used $\sigma$ as an index of stabilization of estimation weights within each participant; the smaller the value of $\sigma$, the more stable the psychophysical function in the phase.

Figure 1e shows that $\sigma$ decreased significantly from Phase 1 to Phase 3 in the pair condition [paired $t$ test: $t_{(41)} = 34.86$, $P < 0.001$, Cohen's $d = 5.38$], but increased in the individual condition [paired $t$ test: $t_{(20)} = -31.37$, $P < 0.001$, Cohen's $d = 6.85$]. The decrease in $\sigma$ was greater in the pair condition than in the individual condition [Welch's $t$ test: $t_{(42.4)} = 46.02$, $P < 0.001$, Cohen's $d = 12.18$]. These results support H2's claim that pair interaction stabilizes the covert psychophysical function within each individual. Furthermore, participants whose estimation stabilized more from Phase 1 to Phase 3 (greater decrease in $\sigma$) also showed greater improvement in accuracy in ordering the stimuli (Supplementary Fig. 3a; robust $r = -0.39$, $P = 0.011$). That is, participants' improvements in post-interaction stability correlated with improvements in rank ordering of the stimuli.

To identify what features of interaction facilitated the covert-level convergence and the stabilization of individual psychophysical functions afterward, we conducted an fMRI experiment and manipulated the paired interaction patterns. To reduce the risk of reverse inference[62], we defined the regions of interest (ROIs) in the mentalizing network a priori for each participant using a functional localizer for perspective taking[59,63,64], separately from the main task (see Methods section for details).

**fMRI Experiment**. The experiment consisted of one pre-interaction phase, two interaction phases, and two post-interaction phases (Fig. 2a). In the pre-interaction phase, participants performed the dot-estimation task individually (Supplementary Fig. 4a).

In the interaction phases, participants were paired with a Sherif- or Asch-type computer partner (described as another participant). Participants and the partner estimated the number of dots independently and then shared their estimates (Supplementary Fig. 4b). Although both computer partners had a stronger underestimation bias initially ($w = 0.61$) than the average participant in the behavioral experiment (mean $w = 0.91$; Fig. 1b), they were programmed to differ in their reciprocity to participants' estimates during interaction (see Methods section for details).

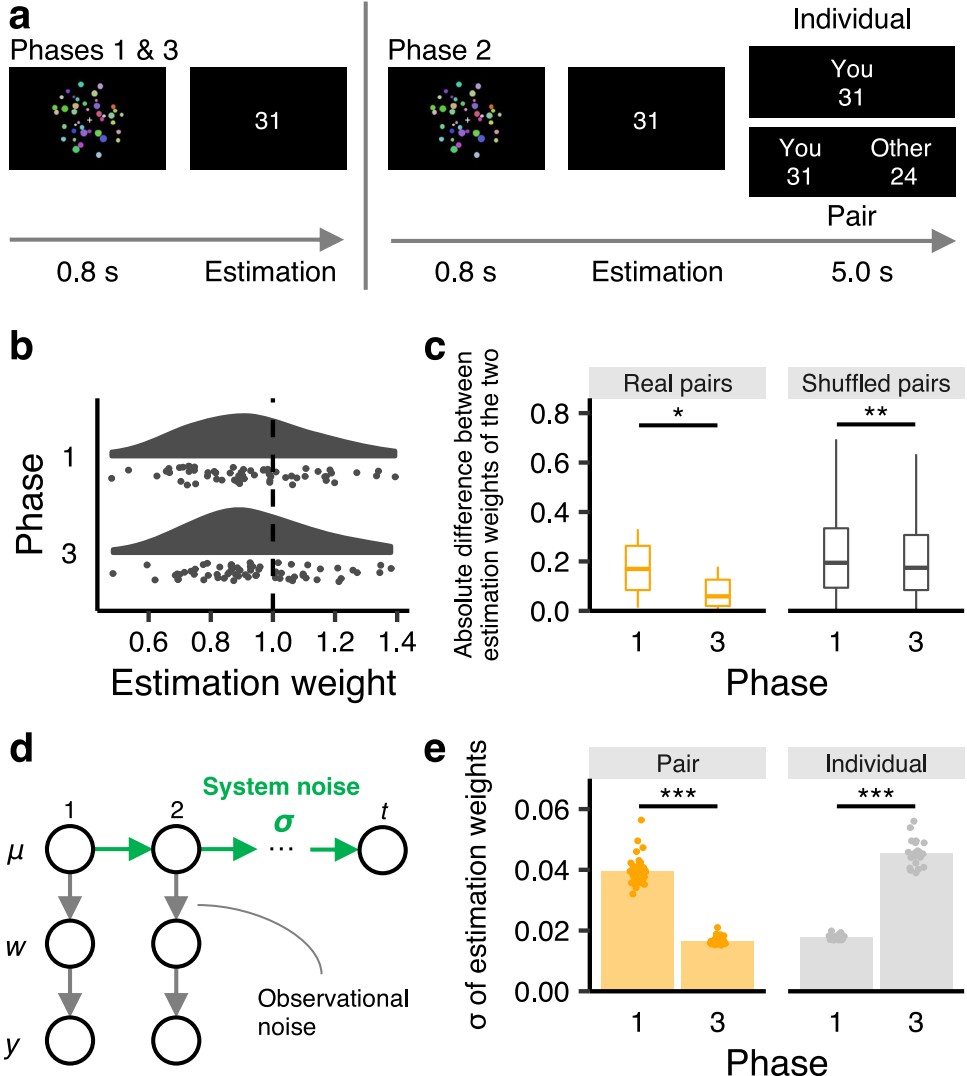

**Fig. 1 Task and results from the behavioral experiment. a** Timelines of the dot-estimation task. In Phases 1 and 3 (solo phases), all participants estimated the number of dots individually. In Phase 2, participants in the individual condition ($n = 21$) performed the task individually again as a control. Those in the pair condition ($n = 42$) observed the same dots and estimated the number of dots independently, and then their estimates were shared with each other. **b** Participants' estimation weights ($w_i$) in Phases 1 and 3 in the pair condition. Each point indicates one participant's estimation weight. **c** Absolute differences in estimation weights between two participants of the real and shuffled pairs in Phases 1 and 3. The box plots indicate the medians, the first and third quartiles, and the values no further than 1.5 inter-quartile range from the quartiles. $*P < 0.05$, $**P < 0.01$. **d** An illustration of the state-space model that was used to measure the stability of estimation weights within each individual. We assumed that the estimation weight at trial $t$, $w(t)$, was sampled with observational noise from a latent state, $\mu(t)$. The latent state $\mu(t)$ was in turn assumed to be sampled with system noise ($\sigma$) from the state in the preceding trial, $\mu(t\text{-}1)$. We used $\sigma$ as an index of stabilization of estimation weights within each participant. In the bottom row, $y$ indicates the participant's estimate. **e** Sigmas of estimation weights in the pair and individual conditions. The bar plots indicate the means across participants. $***P < 0.001$.

The Asch-type partner retained its underestimation bias throughout the trials irrespective of the participant's estimates[26] (Eq. 8). In contrast, the Sherif-type partner adjusted its weight over the trials toward the participant's estimates[18]. The Sherif-type partner was designed to approximate the pattern of an average participant's estimation-adjustment during interaction in the behavioral experiment (Eqs. 9 and 10).

In the post-interaction phases, participants performed the task individually again. Hereafter, each post-interaction phase is referred to as post-Sherif or post-Asch.

**Reciprocal concession stabilizes the covert psychophysical function within each individual after interaction (H3).** For both partners, participants' estimation weights decreased after interaction

[Fig. 2b; post-Sherif: paired $t$ test: $t_{(27)} = 2.35$, Holm–Bonferroni-corrected $P = 0.053$, Cohen's $d = 0.44$; post-Asch: paired $t$ test: $t_{(27)} = 3.87$, Holm–Bonferroni-corrected $P = 0.002$, Cohen's $d = 0.73$]. No significant difference was observed between the partners [paired $t$ test: $t_{(27)} = 1.55$, Holm–Bonferroni-corrected $P = 0.132$, Cohen's $d = 0.29$]. These results indicate that participants were influenced by both partners and increased their underestimation biases through interaction.

However, as expected, the Sherif- and Asch-type partners had different impacts on the stability of the covert psychophysical function for each individual (indexed by the $\sigma$ of estimation weights, as in Fig. 1d). Individual estimation weights were more stable after interaction with both partners [Fig. 2c; reciprocating Sherif-type partner: paired $t$ test: $t_{(27)} = 33.78$, Holm–Bonferroni-corrected $P < 0.001$, Cohen's $d = 6.38$; non-reciprocating Asch-type

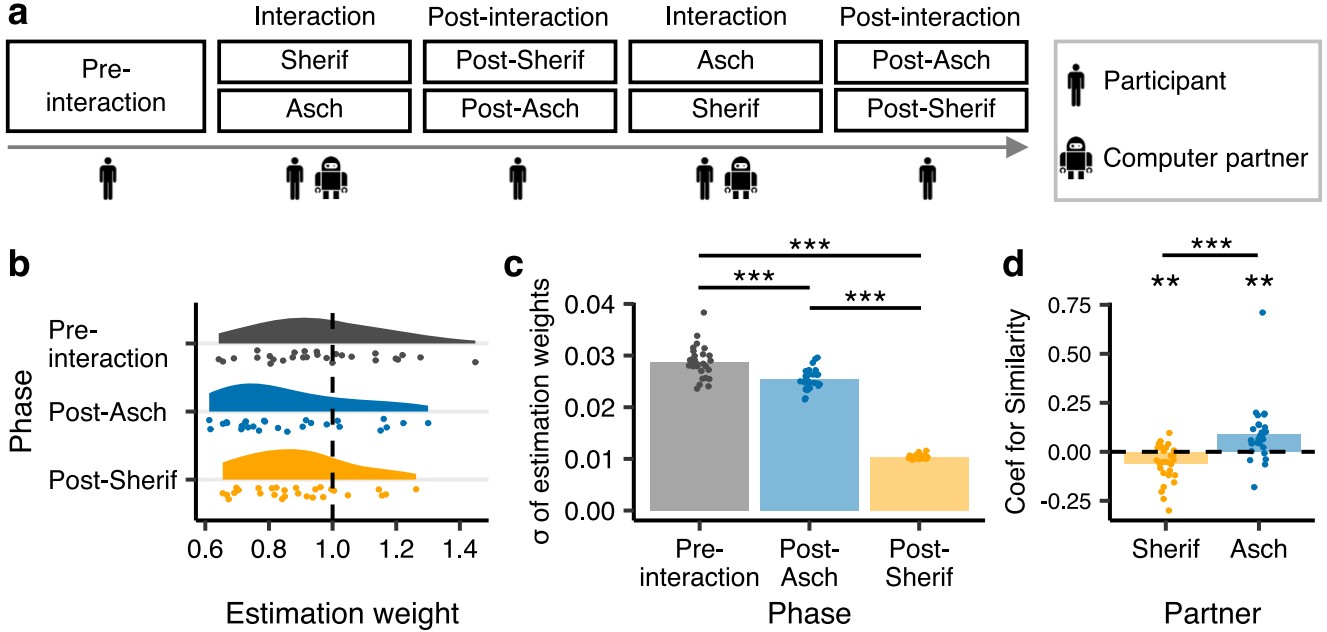

**Fig. 2 Task flow and behavioral results from the functional magnetic resonance imaging experiment. a** The flow of the fMRI experiment. In the pre- and post-interaction phases, participants performed the dot-estimation task individually. In each interaction phase, the participant was paired with the Sherif- or Asch-type partner that had a much stronger underestimation bias initially ($w = 0.61$) than the average participant of the behavioral experiment. The order of the two partners was counterbalanced across participants. The participant and the partner observed the same dots and estimated the number of dots independently, and then their estimates were shared. **b** Participants' estimation weights ($w_i$) in each phase. Each point indicates one participant's estimation weight. **c** Sigmas of estimation weights, indexing stability of the participant's covert psychophysical function. ***$P < 0.001$. **d** Coefficients for similarity (Sim) for the two computer partners in the time-series analysis of participants' estimates during interaction. **$P < 0.01$, ***$P < 0.001$. The bar plots indicate the means across participants.

partner: paired $t$ test: $t_{(27)} = 4.56$, Holm–Bonferroni-corrected $P < 0.001$, Cohen's $d = 0.86$]. Nevertheless, the impacts between the partners are distinct [Fig. 2c; paired $t$ test: $t_{(27)} = 39.56$, Holm–Bonferroni-corrected $P < 0.001$, Cohen's $d = 7.48$]. These results support H3. Reciprocal concession, as implemented with the Sherif-type partner, was more effective for improving the stability of covert psychophysical functions that generate overt behaviors (see Supplementary Fig. 3b for individual-level correlations between improvements in estimation stability and in relative ordering of the stimuli).

**Time-series analysis of updating the estimation weight during interaction.** To examine the cognitive dynamics underlying the different impacts of the Sherif- and Asch-type partners (Fig. 2c) at a finer time resolution, we analyzed how participants updated their estimation weights during interaction using a time-series model (see Methods section and Eqs. 11–13 for details; see Supplementary Figs. 5 and 6 for model validation). The model decomposed a participant's estimation weight ($w_i$ in Eq. 1) in trial $t$ into three parameters:

$$w_i(t) = Baseline + Coef_{Atyp}(t) \times Atyp(t-1) + Coef_{Sim}(t) \times Sim(t-1), \tag{2}$$

where $Baseline$ (i.e., intercept) indicates the participant's estimate without social influence, and $Atyp$ (atypicality) is a nuisance parameter that controls for the built-in underestimation bias of the computer partner (Eq. 12).

$Sim$ (similarity) is a pair-level parameter, representing closeness between the participant's estimate and the partner's estimate in the preceding trial (Eq. 13). As we are concerned with reciprocal concession over time (H3), $Sim$ is the parameter of primary interest. The coefficient for $Sim$ ($Coef_{Sim}$) indicates how the participant updates his/her estimation weight in the current trial in response to

the similarity in the preceding trial. Since $Sim$ is positive and the computer partner initially had a stronger underestimation bias ($w = 0.61$) than the average participant ($w = 0.91$), a negative coefficient means that the participant approaches the partner (i.e., decreases his/her estimation weight) in the current trial from his/her baseline, whereas a positive coefficient means the opposite shift. (Note that Eq. 2 focuses on the participant's weight adjustment from his/her baseline during interaction. We examined whether greater estimation similarity in the preceding trial would generally increase the participant's underestimation bias in the current trial to approach the computer partner, which was endowed with a strong underestimation bias.)

Figure 2d shows that the mean coefficient for similarity was negative when participants interacted with the Sherif-type partner [$M = -0.06$, one-sample $t$ test: $t_{(27)} = 3.46$, $P = 0.002$, Cohen's $d = 0.65$], and positive when interacting with the Asch-type partner [$M = 0.09$, one-sample $t$ test: $t_{(27)} = 3.19$, $P = 0.004$, Cohen's $d = 0.60$]. The mean coefficients were significantly different between the partners [paired $t$ test: $t_{(27)} = 5.46$, $P < 0.001$, Cohen's $d = 1.03$].

These patterns are in line with H3. When interacting with the Sherif-type partner, larger similarity in the preceding trial fostered more similarity in the current trial. Participants reciprocated concession when interacting with the reciprocating Sherif-type partner. In contrast, when interacting with the Asch-type parnter, participants moved away from the partner in response to the preceding similarity.

**Activity of the mentalizing network during bilateral influence modulates stabilization of an individual's covert psychophysical function after interaction (H4).** To test whether perspective taking underpins the differing impacts between the Sherif- and Asch-type partners (Fig. 2c, d), we focused on the mentalizing network, which

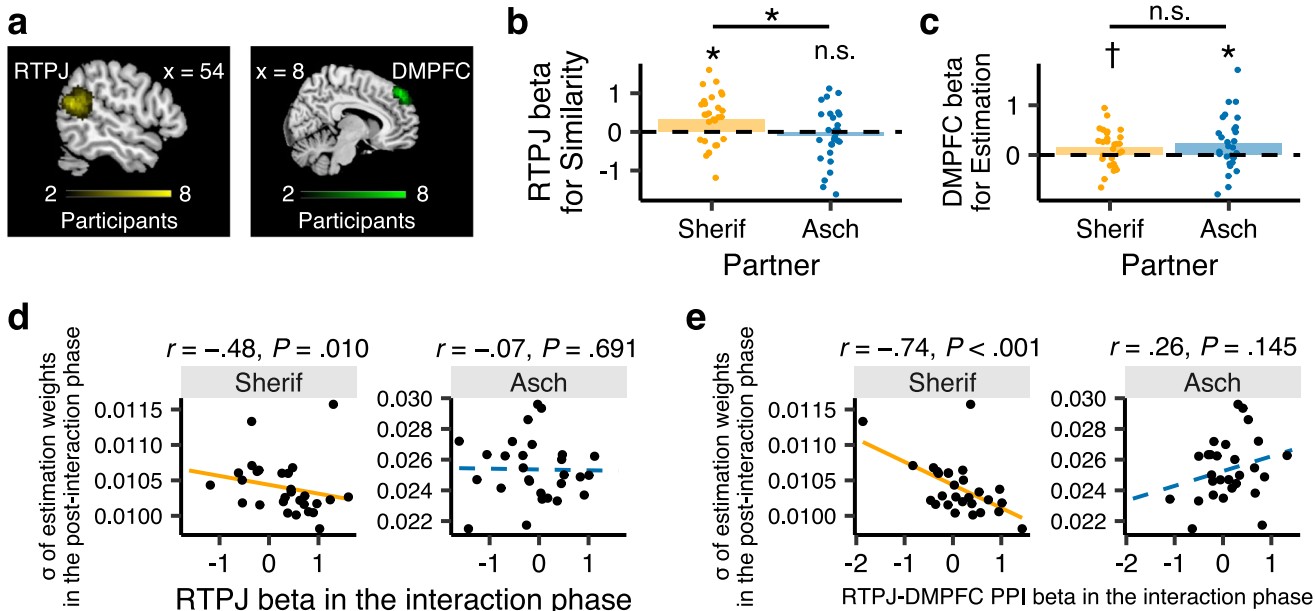

**Fig. 3 Imaging results from the functional magnetic resonance imaging experiment. a** Individual right temporoparietal junction (RTPJ) and dorsomedial prefrontal cortex (DMPFC) regions of interest (ROIs) identified a priori by the functional localizer for cognitive perspective taking based on a theory-of-mind task, conducted separately from the main estimation task. The color bars show the number of overlapped individual ROIs in each voxel. **b** Parametric modulation for *Sim* (similarity) with the partner in estimation (Eq. 13) in RTPJ activity during interaction. *$P < 0.05$. **c** Parametric modulation for *Est* (estimation: Eq. 11) in DMPFC activity during interaction, with larger activation for larger estimates. †$P < 0.10$, *$P < 0.05$. **d** Correlations between RTPJ betas (in response to *Sim* during interaction: Fig. 3b) and the stability ($\sigma$) of the participants' estimation weights in the post-interaction phase. **e** Correlations between RTPJ–DMPFC functional connectivity during interaction and the stability ($\sigma$) of the participants' estimation weights in the post-interaction phase. The bar plots indicate the means across participants, and the lines in the scatter plots are the linear regression lines.

includes the RTPJ and the DMPFC. Using the individual ROIs associated with cognitive perspective taking[65] (Fig. 3a, Supplementary Fig. 7, and Supplementary Table 2), we compared activities of the RTPJ and DMPFC for interactions with the two computer partners.

At the group level (Fig. 3b), RTPJ activity tracked similarity with the partner in estimation in the preceding trial (Eqs. 11 and 13) when participants interacted with the Sherif-type partner [one-sample *t* test: $t_{(27)} = 2.63$, $P = 0.014$, Cohen's $d = 0.50$] but not with the Asch-type partner [one-sample *t* test: $t_{(27)} = 0.69$, $P = 0.495$, Cohen's $d = 0.13$]. The difference between the two types was significant [paired *t* test: $t_{(27)} = 2.25$, $P = 0.033$, Cohen's $d = 0.42$]. These differential RTPJ activities are in line with participants' trial-by-trial updating of estimation weights in response to similarity (Fig. 2d). This may suggest that participants engaged in perspective taking with the Sherif-type partner according to the degree of reciprocal concession during interaction, but not as much with the Asch-type partner. Figure 3c shows that the DMPFC tracked participant behavioral output in the current trial [$Est_i(t)$ in Eq. 11] for both partners [Sherif-type: one-sample *t* test: $t_{(27)} = 2.05$, $P = 0.050$, Cohen's $d = 0.39$; Asch-type: one-sample *t* test: $t_{(27)} = 2.29$, $P = 0.030$, Cohen's $d = 0.43$], with larger activation for larger estimates. Participants' estimation outputs were similarly represented by the DMPFC activity across the two types [paired *t* test: $t_{(27)} = 0.68$, $P = 0.502$, Cohen's $d = 0.13$].

At the individual level (Fig. 3d), RTPJ activity in response to *Sim* during interaction (i.e., RTPJ beta in Fig. 3b) correlated with stability of estimation ($\sigma$) in the post-Sherif phase [robust $r = -0.48$, $P = 0.010$], but not in the post-Asch phase [robust $r = -0.07$, $P = 0.691$]. We also examined the contribution of RTPJ–DMPFC functional connectivity to the stability of estimation using a psychophysiological interaction analysis (see Methods section). Figure 3e shows that RTPJ–DMPFC functional connectivity during interaction correlated with stability in the post-Sherif

phase [robust $r = -0.74$, $P < 0.001$] but not in the post-Asch phase [robust $r = 0.26$, $P = 0.145$].

These results support H4. At the group level, the RTPJ was activated according to similarity (i.e., closeness to estimates in the preceding trial: Eq. 13) when participants interacted with the reciprocating Sherif-type partner, but not with the non-reciprocating Asch-type partner (Fig. 3b). At the individual level, such cognitive perspective-taking toward the reciprocating partner was correlated with the stability (reliability) of the participant's own covert psychophysical function after the interaction. These results imply that participants took the Sherif-type partner's perspective according to the degree of reciprocity when making their own estimates during interaction and stabilized the covert psychophysical function that generated their responses (Fig. 3d, e; see Supplementary Note 1 and Supplementary Fig. 8 for the results of an exploratory whole-brain analysis; see also Supplementary Notes 2 and 3 and Supplementary Figs. 9–11 for other behavioral results).

**Online behavioral experiment**. We conducted a pre-registered experiment to replicate the behavioral results from the fMRI experiment as well as examine whether similar behavior could be observed when the computer partner is an overestimator. In the pre- and post-interaction phases, participants performed the task individually. In the interaction phase, participants and the computer partner performed the task as in the fMRI experiment (Supplementary Fig. 4b). For the computer partner, we had a $2 \times 2$ between-participants design, with factors partner type (Sherif vs. Asch) and built-in estimation bias (underestimation vs. overestimation; see Methods section for details).

**Interaction with the Sherif-type partner stabilizes the covert psychophysical function in both estimation-bias conditions.** Participants decreased their estimation weights after interacting

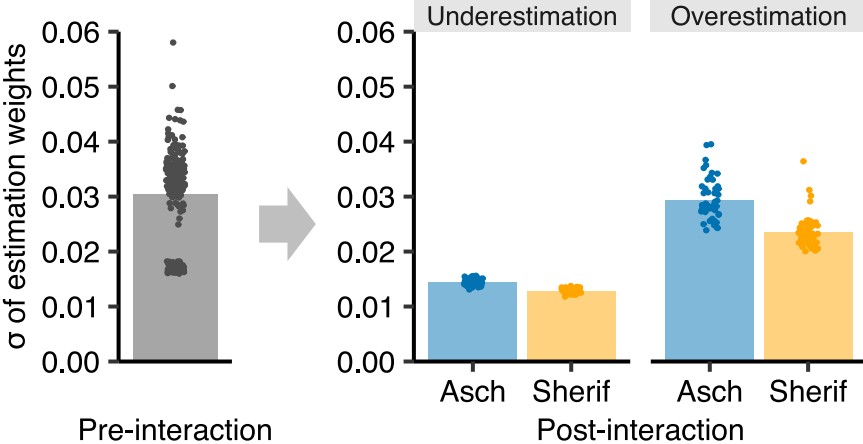

**Fig. 4 Stabilization of estimation weights after interaction in the online behavioral experiment.** Regardless of the estimation biases, participants' estimation weights became more stable (i.e., smaller σ of estimation weights) after interacting with the Sherif-type partner than with the Asch-type partner. The effect of the Sherif-type partner was more pronounced in the overestimation condition than in the underestimation condition. The bar plots indicate the means across participants.

with the underestimators [Supplementary Fig. 12; Asch-type partner: paired $t$ test: $t_{(59)} = 8.65$, Holm–Bonferroni-corrected $P < 0.001$, Cohen's $d = 1.12$; Sherif-type partner: paired $t$ test: $t_{(48)} = 2.49$, Holm–Bonferroni-corrected $P = 0.033$, Cohen's $d = 0.36$]. This pattern replicated the results of the fMRI experiment (Fig. 2b). When interacting with the overestimators, participants increased their estimation weights for the Asch-type partner [paired $t$ test: $t_{(50)} = 5.82$, Holm–Bonferroni corrected $P < 0.001$, Cohen's $d = 0.82$] but not for the Sherif-type partner [paired $t$ test: $t_{(55)} = 1.95$, Holm–Bonferroni corrected $P = 0.056$, Cohen's $d = 0.26$]. A 2 × 2 (partner type × estimation bias) analysis of variance (ANOVA) yielded a significant interaction [$F_{(1, 212)} = 33.09$, $P < 0.001$, $\eta_p^2 = 0.14$] and a main effect of the estimation bias [$F_{(1, 212)} = 53.61$, $P < 0.001$, $\eta_p^2 = 0.20$]. These patterns indicate that participants changed their estimation weights toward the computer partners, except for the overestimating Sherif-type that approached participants' estimates rapidly during early phases of the interaction, as programmed (see Methods section and Eq. 10).

We next examined whether the results on increased estimation stability (Fig. 2c) are replicable. Figure 4 displays σs of participants' estimation weights. A 2 × 2 (partner type × estimation bias) ANOVA on the pre–post differences yielded significant main effects for both factors [partner type: $F_{(1, 212)} = 388.03$, $P < 0.001$, $\eta_p^2 = 0.65$; estimation bias: $F_{(1, 212)} = 1373.32$, $P < 0.001$, $\eta_p^2 = 0.87$]. This indicates that σ decreased more after interaction with the Sherif-type partner than with the Asch-type partner (H3) and decreased more after interaction with the underestimator than with the overestimator. The interaction effect was also significant [$F_{(1,212)} = 410.39$, $P < 0.001$, $\eta_p^2 = 0.66$], suggesting that the increased stability when interacting with the reciprocating Sherif-type partner (compared to the Asch-type) was more pronounced in the overestimator condition. These results replicate and extend the behavioral results of the fMRI experiment. Reciprocal concession stabilized participants' covert psychophysical functions in both estimation-bias conditions (H3). The individual-level correlations between improvement in estimation stability and improvement in relative ordering of the stimuli from the pre- to post-interaction phase were also replicated (Supplementary Fig. 3c).

## Discussion

We have addressed the neuro-cognitive underpinnings of perceptual norms as an emergent property via dynamic interaction, using a dot-estimation task. We hypothesized that people's

perceptual responses are fined-tuned as shared realities through bilateral (more than unilateral) influence and perspective taking and tested this claim using behavioral (laboratory and online) and fMRI experiments.

The results support our hypotheses. In the laboratory behavioral experiment using real pairs, dyadic interaction yielded pairwise convergence of not only participants' overt behaviors but also their covert psychophysical functions (H1: Fig. 1c). The paired interaction also stabilized the psychophysical function within each individual as an aftereffect (H2: Fig. 1e).

In the fMRI experiment, we created two computer partners: a reciprocating Sherif-type (similar to most participants in the behavioral experiment) and a non-reciprocating Asch-type, both endowed with a much stronger underestimation bias than the average participant. At the behavioral level, both computer partners increased participants' mean underestimation bias as an aftereffect. Notice that, similar to the original Sherif task[18], our task contained greater perceptual ambiguity than the original Asch task (comparing the length of lines), which may have contributed to the impact of both the Asch- and Sherif-type partners on participants' estimations after interaction in our study (Fig. 2b and Supplementary Fig. 12). On the other hand, as hypothesized (H3), within-individual stability of the covert psychophysical function improved more after interacting with the Sherif-type compared to the Asch-type partner (Fig. 2c). Furthermore, the time-series analysis confirmed that participants adjusted their estimation weights reciprocally during interaction (i.e., approaching the partner's estimate more closely in the current trial in response to the similarity in the preceding trial) only with the Sherif-type partner (Fig. 2d).

At the neural level, the mentalizing network (defined a priori using a functional localizer based on a theory of mind task[59,63,64]) contributed to the dynamic formation and stabilization of perceptual norms. First, consistent with the behavioral results (Fig. 2d), RTPJ activity tracked temporal changes in estimation similarity during interaction, when paired with the reciprocating Sherif-type but not with the one-sided Asch-type partner (Fig. 3b). Such RTPJ activity, which parametrically modulated the estimation similarity with the Sherif-type partner on a trial-by-trial basis, also contributed to the subsequent stabilization of participants' own covert psychophysical functions (H4: Fig. 3d). Second, we observed that DMPFC activity tracked how participants expressed their estimations as outputs during interaction (Fig. 3c). The involvement of the DMPFC in expressing estimations was observed commonly for participants'

interactions with both Asch- and Sherif-type partners (with larger activation when giving larger estimates). It might be the case that expressing a larger estimation in the presence of a partner with a strong underestimation bias was a socially nuanced behavior, which seems to concur with the previous finding that the DMPFC modulates self-related evaluation[66] embedded in social contexts[67–69]. Although the RTPJ and the DMPFC are the core areas of the mentalizing network, the functional dissociation of these areas has been proposed[70]: Whereas the RTPJ is thought to facilitate perspective taking, the DMPFC is thought to integrate social information into subjective evaluation[71–73]. Our results appear to be in line with this view. Last, the functional connectivity between the RTPJ and the DMPFC modulated the stabilization of participants' covert psychophysical functions after interacting with the reciprocating Sherif-type but not with the unresponsive Asch-type partner (H4: Fig. 3e). These regions are known to be recruited even when participants are not explicitly instructed to infer another's mental state[58,74–76]. Our participants were also instructed that they did not need to give similar estimates and that their reward would depend entirely on their own estimation accuracy. Thus, the fMRI results indicate that participants spontaneously had the reciprocating partner in mind when they made their own estimations during as well as after interaction.

Furthermore, in the pre-registered online behavioral experiment, we replicated and extended the behavioral results of the fMRI experiment. In addition to showing the impact of the Asch- and Sherif-type partners on participants' estimations after interaction (Supplementary Fig. 12), the follow-up experiment revealed that reciprocal concession (as implemented in the Sherif-type partner) particularly stabilized participants' covert psychophysical functions, not only in the underestimation condition but also in the overestimation condition (Fig. 4). These patterns provide further support for H3.

Taken together, the results suggest that reciprocal behavioral concession may lead people to take the perspective of their interaction partners[59,63,65]. Such spontaneous switching between one's own and another's view about shared targets may stabilize people's covert psychophysical functions, as triangulation measurement is useful for reliably determining the location of a point.

We believe that these results are important in two respects. First, in the digital era, new shared realities can emerge rapidly and robustly and often fundamentally contradict democratic values (e.g., conspiracy theories, deliberate misinformation). Research on formation of new norms (as opposed to acquisition of extant norms) is thus of keen importance. Echterhoff et al. (2011)[77] and Higgins et al. (2021)[78] argued that shared reality is not a mere duplication or catching of another person's response but requires that one's inner state about some target referent converges with the other's inner state regarding the same target. Our finding that bilateral influence causes convergence of not only overt responses about the same perceptual stimuli but also covert psychophysical functions via perspective taking fits this view. It may also be noteworthy that participants subjectively reported having felt their inner (generative) model converged more with that of the Sherif-type partner than with the Asch-type partner (Supplementary Fig. 11), but they were actually influenced to the same extent by both types (Fig. 2b). Along with the increased stability of the psychophysical function after interaction (Figs. 2c and 4), these results indicate that reciprocal social influence[35] played a key role in fostering the shared reality in our experiment (see Echterhoff et al.[77] and Higgins et al.[78] for further discussion on shared realities). It is also noteworthy that a shared stable psychophysical function allows people to agree on a totally new target (not in the original learning set) immediately. These generalized endogenous agreements may further foster a shared

sense of feeling and thinking the same way. We believe that such a shared generative model[33] is one of the most fundamental characteristics of social norms[79,80].

Second, our results appear to shed light on overlooked aspects of collective decision-making and its accuracy (the wisdom of crowds or WoC[15,81]). Previous research on the WoC has yielded mixed views regarding the effect of interaction, with some arguing negative impacts[35,82] and others reporting positive impacts[60]. Moreover, although most WoC research has focused on improved accuracy in aggregated numeric estimates[35,83,84], aggregating ordered choices has also been shown to yield a WoC effect[34,85]. The present study has shown that, although interaction did not necessarily correct people's underestimation bias (Figs. 1b, 2b, and Supplementary Fig. 12), bilateral influence stabilized their covert psychophysical functions more than unilateral influence (Figs. 2c and 4). Recall that, as illustrated with the weighing metaphor, an increase in stability (reliability) of the psychophysical function can improve the accuracy of ordering of estimation targets (Supplementary Fig. 3). In other words, even though people remain biased numerically, bilateral influence may improve individual ordering of the targets over time and enhance accuracy of group choice accordingly (e.g., aggregated ordering by majority/plurality voting[34,85]). Investigating collective decision-making accuracy from this perspective appears to be a promising approach[15].

People rely on constant interaction with others to develop and sustain shared realities[1,2]. Exploring the applicability of the computational approach (used here to study perceptual norms[18]) to more social norms (about shared values and morals[86]), while focusing on the formation of shared generative models[33], could be a promising way to better understand emerging problems in our digitally connected but often morally divided world.

## Methods
This study was approved by the Ethics Committee of the Department of Social Psychology at the University of Tokyo, and all participants gave written informed consent.

**Participants**. Sixty-three students at the University of Tokyo (pair condition: 27 men and 15 women; age 21.2 ± 1.0 years; individual condition: 14 men and 7 women; age 22.0 ± 1.7 years) participated in the laboratory behavioral experiment. The sample size was determined by a power analysis on findings from a previous study employing a similar experimental paradigm[87]. For the functional magnetic resonance imaging (fMRI) experiment, we scanned 30 right-handed students at the University of Tokyo (who did not overlap with participants in the behavioral experiment) with no history of neurological or psychiatric illness. Before analyzing the data, we excluded two participants who did not follow the experiment instructions, which left us a total of 28 participants (13 men and 15 women; age 22.2 ± 2.3 years) for analysis. For the online behavioral experiment, we recruited 216 students at the University of Tokyo who had not participated in the previous experiments (132 men and 84 women; age 23.0 ± 2.1 years). All participants had normal or corrected-to-normal visual acuity and no color vision deficiency.

**Laboratory behavioral experiment**. In the laboratory behavioral experiment (Fig. 1a), participants were presented with 25–55 randomly sized and colored dots for 0.8 s (the luminance of each screen was held constant), and then asked to estimate the number of dots in each trial using a keypad, with no time limit. The number of dots ranged from 25 to 55 in increments of two, and its presentation order was randomized across participants. The intertrial interval was 2 s with a fixation cross. No feedback was given to participants about their estimation accuracy throughout the experiment. We used a laptop computer (HP ZBook 15 Mobile Workstation, HP) and a set of Psychtoolbox-3 scripts[88–90] to control the experiment. Stimuli were presented on a 15.6-inch LCD monitor (1920 × 1080 resolution; MB168B + , AsusTek Computer), 60 cm in front of the participant. Screen luminance was kept constant across trials.

The experiment consisted of three phases, and there were 16 trials each in Phases 1 and 3 for analysis. Phase 2 had 144 trials in order to generate sufficient data to implement average participant behavior as a computer agent later for the fMRI experiment. In Phases 1 and 3, all participants estimated the number of dots individually. In Phase 2, 21 participants performed the task individually again as a control, and the remaining 42 participants performed the task in pairs. Separated

by a partition that blocked any visual contact, each pair observed the same dots and estimated the number of dots individually, and then each individual viewed both their own and their partner's estimates on their own respective monitors for 5 s (Fig. 1a right). Participants remained completely anonymous to each other and were allowed no verbal communication throughout the experiment. After the experiment, participants were paid according to their respective individual performances. The amount of each participant's additional reward was determined by the average absolute error from the actual number of dots. If the average error was < 5, < 10, < 15, < 20, < 25, or > 25, the additional reward was 1000, 800, 600, 400, 200, or 0 Japanese yen, respectively (this method was used in the fMRI and online experiment as well). Each participant's estimation weight ($w_i$ in Eq. 1) was calculated for Phases 1 and 3 (solo phases) separately using the maximum likelihood method.

**Linear versus log-linear models of dot estimation**. We first constructed two simple psychophysical models of participant $i$'s estimated number of dots at trial $t$. One was a linear model (Eq. 1), and the other was a log-linear model:

$$Est_i(t) = w_i \times \log(DotNum(t)) + \varepsilon. \qquad (3)$$

The notation is the same as in Eq. 1. As the linear model outperformed the log-linear model in terms of Akaike information criterion (see Supplementary Table 1), we used the linear model in the analysis.

**Estimating the stability of estimation weights**. To examine whether the interaction stabilized participants' estimation weights, we adopted a hierarchical local-level model, which is a basic form of the state-space modeling approach[61]. This type of model consists of the state equation (capturing the latent dynamics of the system) and the observation equation (transforming the state into the observation). The state-space modeling approach enabled us to assess how estimation weights changed over time, while considering autocorrelation between trials as well as observational random noise in each trial (see Fig. 1d for illustration). We assumed that the estimation weight for individual $i$ ($w_i$) at trial $t$ was sampled with observational noise ($\sigma_{obs,i}$) from a latent state ($\mu_i$). The latent state was in turn assumed to be sampled with system noise ($\sigma_{\mu,i}$) from the state in the preceding trial. We used $\sigma_{\mu,i}$ as an index of stabilization of estimation weights within each participant. In the main text, we refer to $\sigma_{\mu,i}$ as $\sigma$. We assumed that $\sigma_{obs,i}$ and $\sigma_{\mu,i}$ are generated from a normal distribution with group-level mean $g\mu_{obs}$, variance $g\sigma_{obs}$, and mean $g\mu_\mu$, variance $g\sigma_\mu$, respectively.

$$w_i(t) \sim N\left(\mu_i(t), \sigma_{obs,i}\right) \qquad (4)$$

$$\mu_i(t) \sim N\left(\mu_i(t-1), \sigma_{\mu,i}\right) \qquad (5)$$

$$\sigma_{obs,i} \sim N\left(g\mu_{obs}, g\sigma_{obs}\right) \qquad (6)$$

$$\sigma_{\mu,i} \sim N\left(g\mu_\mu, g\sigma_\mu\right) \qquad (7)$$

These parameters were estimated using a Markov chain Monte Carlo (MCMC) method implemented in R (ver. 3.6.1) and rstan (ver. 2.21.2). We used uninformative priors (uniform distributions on the parameters; lower limit $= -\infty$ and upper limit $= \infty$ for $\mu_i$, $g\mu_{obs}$, and $g\mu_\mu$; lower limit $= 0$ and upper limit $= \infty$ for $\sigma_{obs,i}$, $\sigma_{\mu,i}$, $g\sigma_{obs}$, and $g\sigma_\mu$. We generated the posterior distributions of the parameters by MCMC sampling from four chains of iterations while thinning the samples to avoid autocorrelation and obtain 1000 posterior samples each. The first 500 iterations in each chain were discarded as burn-in periods.

**fMRI experiment**. We scanned 28 participants (13 men and 15 women) in the fMRI experiment. Prior to the experiment, we conducted a power analysis assuming a two-tailed paired $t$ test with $\alpha = 0.05$, $\beta = 0.8$, and Cohen's $d = 0.6$, using G*power[91]. We found that 24 or more participants were needed. Participants performed the same dot-estimation task as in the laboratory behavior experiment. The number of dots was varied between 25 and 58 in increments of three, and its presentation order was randomized across participants. Participants were instructed to estimate the number of dots within 5 s. The intertrial interval was 2, 3, or 4 s with a fixation cross. No feedback was given to participants about their estimation accuracy, as in the behavioral experiment. Participants used a response pad (HHSC-1×2-BY, Current Designs) to give their estimate of the number of dots. A workstation (Precision Tower, Dell) controlled the experiment, and stimuli were presented on an MRI-compatible 32-inch LCD monitor (1920 × 1080 resolution; NNL-LCD, NordicNeuroLab) behind the MR bore. An eye tracker (EyeLink 1000, SR Research) recorded participants' eye movements at 250 Hz. The experiment consisted of one pre-interaction phase, two interaction phases, and two post-interaction phases (24, 48, and 24 trials, respectively: Fig. 2a and Supplementary Fig. 4). In the pre- and post-interaction phases, each participant performed the dot-estimation task individually. In each interaction phase, the participant was paired with the Sherif- or Asch-type computer partner, which was described as another participant. The participant and the computer partner estimated the number of dots independently, and then both estimates were shown

after every trial. The Sherif-type partner adjusted its estimation weights in response to the participant during interaction, whereas the Asch-type partner remained unaffected by the participant's estimates throughout the phase. Both computer partners had a strong built-in underestimation bias initially, compared to the average participant of the laboratory behavioral experiment.

To identify individual ROIs in the RTPJ and the DMPFC a priori, we conducted a localizer task for cognitive perspective taking based on theory of mind (ToM)[59,63,64], separately from the main task. The localizer task consisted of seven ToM items and seven non-ToM items. Using the individual ROIs identified by the localizer (Fig. 3a and Supplementary Fig. 7; Supplementary Table 2), we examined whether the RTPJ tracked similarity (Sim) and whether the DMPFC tracked estimation (Est) during interaction. We also examined functional RTPJ–DMPFC connectivity to see whether the RTPJ activity tracking Sim modulated the DMPFC activity for Est.

After scanning, participants answered a post-session questionnaire about the experiment. No participant noted any suspicion about the experimental setting. At the end of the experiment, participants were debriefed and paid according to their performances.

**Data exclusion in the fMRI experiment**. To reduce the effect of outliers, we excluded 11 trials from analysis because the participant's estimate was incomplete or <10. All statistical conclusions were unchanged when we included these trials in the analysis.

**Computer partners**. We devised two computer partners–the Sherif- and Asch-type partners–for the two interaction phases. The partner order was counterbalanced across participants (Fig. 2a). Both partners followed a common estimation model (Eq. 1: Est was rounded to the nearest integer) and had a strong underestimation bias initially ($w = 0.61$, which was 1.5 SD below the mean $w$ of participants in Phase 1 of the laboratory behavioral experiment). The partners differed in behavioral reciprocity to participants' estimates during interaction.

The Asch-type partner was programmed to be unresponsive to the participant's estimate and retained the same estimation weight throughout the interaction. The Asch-type partner estimated the number of dots at trial $t$ as follows:

$$Est_p(t) = 0.61 \times DotNum(t) + \eta, \qquad (8)$$

where $\eta$ denotes the error term, which was sampled uniformly from $\{-2, -1, 0, 1, 2\}$ in every trial. Due to the error term $\eta$, the Asch-type partner appeared to move toward or away from the participant from one trial to the next, which allowed us to do the time-series analysis reported in Eqs. 11–13.

In contrast, the Sherif-type partner was programmed to reciprocate the participant's estimates by updating its estimation weight $w_p(t)$ at trial $t$ as follows:

$$Est_p(t) = w_p(t) \times DotNum(t) + \eta, \qquad (9)$$

$$w_p(t) = 0.61 + \sum_{j=1}^{5} a_j w_p(t-j) + \sum_{j=1}^{5} b_j w_i(t-j), \qquad (10)$$

where $w_p(t-j)$ represents the estimation weight of the Sherif-type partner, and $w_i(t-j)$ represents that of the participant in each of the last five trials preceding the current trial ($j = 1 \dots 5$). Note that 0.61 in Eq. 10 is unrelated to the value of strong underestimation bias, $w = 0.61$.

To simulate an average participant's updating with this approximation model, we first fitted Eq. 10 to the 21 pairs' (i.e., 42 participants') data from the laboratory behavioral experiment. The mean values of estimated $a_j$ ($a_{1\dots5}$) were 0.091, 0.044, 0.037, 0.032, and 0.025 and those of $b_j$ ($b_{1\dots5}$) were 0.066, 0.014, 0.001, −0.015, and −0.011, respectively. We used these values for the Sherif-type partner in Eq. 10.

**Time-series analysis of participants' estimation in the fMRI experiment**. To analyze trajectories of participants' estimation during the course of interaction, we used the following time-series model:

$$Est_i(t) = (Baseline + Coef_{Atyp}(t) \times Atyp(t-1) + Coef_{Sim}(t) \times Sim(t-1))$$
$$\times DotNum(t) + \varepsilon, \qquad (11)$$

where

$$Atyp(t-1) = 1 - \frac{Est_p(t-1)}{0.91 \times DotNum(t-1)}, \qquad (12)$$

$$Sim(t-1) = \frac{\pi}{2} - \tan^{-1} \frac{\left| Est_i(t-1) - Est_p(t-1) \right|}{\left| \left( w_i(Pre) - w_p(Pre) \right) \times DotNum(t-1) \right|}. \qquad (13)$$

The linear structure and notation of Eq. 11 are equivalent to those of Eq. 1. In Eq. 11, however, the participant's estimation weight at trial $t$ ($w_i$ in Eq. 1) was further decomposed into three parameters: Baseline, Atyp (atypicality), and Sim (similarity). Baseline (i.e., intercept) indicates how the participant estimates the number of dots without social influence. Atyp indicates how atypical the partner's estimate was at trial $t$–1. Recall that the Asch-type partner was not only unresponsive to the participant's estimate but also kept its atypical position because

of the initial bias ($w = 0.61$) throughout the interaction. Thus, atypicality is a nuisance parameter of no interest that statistically controls for the effect of the built-in underestimation bias of the computer partner. To quantify the atypicality in Eq. 12, we used the mean estimation weight in the behavioral experiment (0.91) as participants' representative estimate.

*Sim*, which bears on H3 and is of interest here, represents the closeness of the estimates given by the participant ($Est_i$) and by the computer partner ($Est_p$) at trial $t$–1, as compared to their initial closeness prior to interaction. More specifically, as depicted in Eq. 13, it is a pair-level (and thus unsigned) parameter, capturing the degree of closeness between the two estimates at the preceding trial over that before the interaction. The notation $w_i(Pre)$ stands for participant $i$'s estimation weight in the pre-interaction phase, and $w_p(Pre)$ for the computer partner's initial weight (0.61).

To capture temporal changes in the parameters, we adopted a local linear trend model and a state-space modeling approach[61]:

$$Baseline \sim N\left(w_i(Pre), \sigma_{Baseline}\right) \qquad (14)$$

$$Coef_{Sim}(t) \sim N\left(Coef_{Sim}(t-1) + \gamma_{Sim}(t-1), \sigma_{Coef_{Sim}}\right) \qquad (15)$$

$$\gamma_{Sim}(t) \sim N\left(\gamma_{Sim}(t-1), \sigma_{\gamma_{Sim}}\right) \qquad (16)$$

$$Coef_{Atyp}(t) \sim N\left(Coef_{Atyp}(t-1) + \gamma_{Atyp}(t-1), \sigma_{Coef_{Atyp}}\right) \qquad (17)$$

$$\gamma_{Atyp}(t) \sim N\left(\gamma_{Atyp}(t-1), \sigma_{\gamma_{Atyp}}\right) \qquad (18)$$

$$\varepsilon \sim N\left(0, \sigma_\varepsilon\right) \qquad (19)$$

These parameters were estimated using a Markov chain Monte Carlo (MCMC) method implemented in R (ver. 3.6.1) and rstan (ver. 2.17.3). We used uninformative priors (uniform distributions on the parameters; lower limit $= -\infty$ and upper limit $= \infty$ for $Coef_{Sim}$, $Coef_{Atyp}$, $\gamma_{Sim}$, and $\gamma_{Atyp}$; lower limit $= 0$ and upper limit $= \infty$ for $\sigma_{\gamma_{Sim}}$ and $\sigma_{\gamma_{Atyp}}$). We generated the posterior distributions of the parameters by MCMC sampling from four chains of 20,000 iterations each. The first 10,000 iterations were discarded as burn-in periods, and the remainder were thinned by a factor of 50 to avoid autocorrelation.

To confirm model validity, we compared the model with seven reduced models using widely applicable information criteria (WAIC)[92]. The reduced models were (i) Null, (ii) Baseline, (iii) Atypicality, (iv) Similarity, (v) Baseline + Atypicality, (vi) Baseline + Similarity, and (vii) Atypicality + Similarity. We first calculated WAIC for each participant and each interaction phase and then calculated the average WAIC of the two phases for each participant. The full model outperformed the others in terms of WAIC (see Supplementary Fig. 5).

We further performed posterior predictive checking of the model. We generated predictive samples using the means of posterior distributions and compared the actual and predicted estimates of the number of dots for each trial and each participant. Supplementary Fig. 6 clearly shows that there were highly positive correlations between the observed and predicted values in both Sherif and Asch phases (Sherif: $rs > 0.84$; Asch: $rs > 0.89$). These results imply that our model is fitted well to participants' behavior during interaction.

**fMRI data acquisition**. A 3T MR scanner (Prisma, Siemens) was used to acquire magnetic resonance images. A 64-channel head–neck coil was used for radio frequency signal reception. Mild cushioning minimized participant head movement. Heartbeat and respiration were also recorded by a pulse oximeter and pressure sensor attached to the scanner.

Structural images were acquired using a T1-weighted sequence ($1 \times 1 \times 1$ mm³ resolution; 3D-MPRAGE). Functional images were also obtained using a multiband echo-planar imaging sequence provided by the Center for Magnetic Resonance Research, the University of Minnesota[93] (Release R016). The scanning parameters for the echo-planar imaging were as follows: repetition time: 1000 ms; echo time: 30 ms; field of view: $216 \times 216$ mm²; matrix: $72 \times 72$, resolution: $3 \times 3 \times 3$ mm²; 45 slices with no gap; flip angle: 59°; multiband factor[94]: 3. The functional images were taken parallel to the AC–PC line. After discarding the first seven scans to ensure magnetization equilibrium, we acquired 640, 728, 320, and 294 scans in the pre-interaction, interaction, post-interaction, and theory-of-mind localizer task phases, respectively.

**Image analyses**. We used SPM12 (ver. 7219; Wellcome Department of Cognitive Neurology, University College London) working on Matlab R2017a (Mathworks) to process the scanned images. We performed (i) slice-timing correction using the first slice as a reference, (ii) spatial realignment, (iii) co-registration of structural and functional images, (iv) spatial normalization to the Montreal Neurological Institute space, and (v) spatial smoothing (full-width at half-maximum of Gaussian kernel = 8 mm isotropic). Low-frequency noise was removed by a high-pass filter of 128 s.

Three types of general linear models (GLMs) were adopted for each participant in the interaction runs. GLM1 examined the brain regions that tracked similarity in the time-series model (Eq. 13). GLM1 included two condition regressors for cue and feedback as events (see Supplementary Fig. 4b). For the *Cue* regressor, we included five parametric modulators [*Baseline*, $Coef_{Sim}(t)$, $Sim(t-1)$, $Coef_{Atyp}(t)$, and $Atyp(t-1)$], which were derived from the time-series model for each participant, in the design matrix. For the *Feedback* regressor, two parametric modulators [$Sim(t)$, $Atyp(t)$] were included. Each parametric modulator was scaled by z-score normalization. GLM2 examined the functional connectivity related to similarity. GLM2 also included two condition regressors for cue and feedback as events. For the *Cue* regressor, the *Similarity* [$Sim(t-1)$] regressor was included. GLM3 included two condition regressors for cue and feedback as events. For the *Cue* regressor, the estimated dot number for trial $t$ was included.

To remove the effects of head movement and physiological noise, we included nine nuisance regressors (translations along the $x$, $y$, and $z$ axes, rotations of pitch, roll, and yaw, heart rate, respiration, and DVARS) in each GLM (see Power et al.[95] for details of DVARS). To construct a heart rate regressor, we identified the peaks in the 6-s window in the pulse wave signal and obtained the inverse number of the average peak-to-peak duration. To construct a respiratory regressor, we calculated the standard deviation of respiration signal in the 6-s window for each TR.

We first defined the regions of interest (ROIs) for the right temporoparietal junction (RTPJ) and dorsomedial prefrontal cortex (DMPFC) for each participant using the functional localizer task (Fig. 3a and Supplementary Fig. 7). The local peak nearest to the group peak (Supplementary Table 2) was identified individually for all participants for the RTPJ ROI and for 26 of 28 participants for the DMPFC ROI (we substituted the group peak of the DMPFC for the remaining two participants). Both ROIs for each participant were a 6-mm radius sphere centered on the respective individual peak defined using MarsBaR toolbox[96] (ver. 0.44). Using these ROIs, we examined whether RTPJ activity tracked similarity (Eq. 13) while participants interacted with the two computer partners (see Fig. 3b for the results) using GLM1. We also examined whether activity in the DMPFC tracked estimation (the left side of Eq. 11) during interaction phases using GLM3 (Fig. 3c).

**Functional connectivity analyses**. We conducted a generalized form of context-dependent psychophysiological interaction (gPPI) analysis to examine whether the RTPJ activity tracking Similarity modulated the DMPFC activity. Individual ROIs in the RTPJ (Fig. 3a, left) and in the DMPFC (Fig. 3a, right) were defined as described above, with the RTPJ as the seed region and the DMPFC as the target region. The model to calculate gPPI consisted of five factors: seed time series; task regressors {*Cue*, *Feedback*}; products of task regressors and seed time series; the product of *Cue* and parametric modulator *Similarity*; and the product of *Cue*, *Similarity*, and seed time series. Using the MarsBaR toolbox, we extracted the beta coefficient of the product of *Cue*, *Similarity*, and seed time series and examined whether the coefficients predicted the stability of the participants' estimation in the post-interaction phases.

**Functional localizer task**. To identify the brain regions associated with cognitive perspective taking, we used a Japanese version of a mentalizing localizer task[63,97,98]. Participants performed the localizer task after the first interaction phase.

The localizer task consisted of seven theory-of-mind (ToM) items and seven non-ToM items. The ToM items required participants to infer characters' false beliefs, whereas the non-ToM items demanded no such inferences but the same level of logical complexity. In each trial, a short scenario (e.g., "On the morning of the high school reunion, Yuki placed her high-heeled shoes under her dress and then went shopping. That afternoon, her sister borrowed the shoes and later put them under Yuki's bed.") was presented for 10 s, followed by a sentence about the scenario (e.g., "Yuki gets ready assuming her shoes are under the dress."). Participants were asked to choose whether the sentence was true or false within 4 s using the response pad.

The GLM for the mentalizing localizer included ToM and non-ToM as condition regressors, and the duration was set to 14 s. All trials were included in the analysis irrespective of whether the response was correct. The contrast of ToM and non-ToM showed the activation of the RTPJ (Supplementary Fig. 7) and other regions associated with ToM (Supplementary Table 2). We used the peaks of RTPJ activation to define the individual ROI.

**Online behavioral experiment**. In the pre-registered online behavioral experiment (https://osf.io/d2f73/), 216 participants visited the website for the experiment on their computers and were asked to complete the dot-estimation task using a keyboard (Supplementary Fig. 4). As in the laboratory behavioral experiment, the sample size was determined by a power analysis on findings from a previous study employing a similar experimental paradigm[87]. The number of dots was randomly varied across trials in the same way as in the fMRI experiment. However, unlike in the fMRI experiment, no time limit was set for the estimation of the number of dots, to avoid missing data. No feedback was given to participants about their

estimation accuracy, as in the other experiments. The experiment was performed using jsPsych[99].

The experiment consisted of one pre-interaction phase, one interaction phase, and one post-interaction phase (24, 48, and 24 trials, respectively). In the pre- and post-interaction phases, each participant performed the task individually. In the interaction phase, the participant was paired with a computer partner, which was described as another participant. The participant and the computer partner estimated the number of dots independently, and then both estimates were shown after every trial. For the computer partner, we had a $2 \times 2$ between-participant design, with factors partner type (Sherif-type vs. Asch-type) and built-in estimation bias (underestimation vs. overestimation). The response patterns of the Sherif- and Asch-type partners were created using the same algorithms as in the fMRI experiment (Eqs. 8–10). The partners' underestimation and overestimation biases were set to $w = 0.61$ and $1.21$ respectively, which were 1.5 SD below or above the mean $w$ of participants in Phase 1 of the laboratory behavioral experiment (Fig. 1b). Participants were randomly assigned to one of the four conditions (Sherif × Underestimation: $n = 49$; Sherif × Overestimation: $n = 56$; Asch × Underestimation: $n = 60$; Asch × Overestimation: $n = 51$). At the end of the experiment, participants were debriefed and compensated with Amazon gift cards according to their performances. Each participant's estimation weight ($w_i$ in Eq. 1) was calculated for the pre- and post-interaction phases separately using the maximum likelihood method.

**Statistics and reproducibility**. Brain images were pre-processed and analyzed by SPM12 (ver. 7219) working on Matlab R2017a. Statistical analysis was conducted using two-tailed paired and Welch's t-tests and two-way ANOVA using R and rstatix package. Multiple comparison correction was conducted by adjusting p-values according to Holm–Bonferroni method. Correlation between variables were analyzed based on robust statistics with MM-estimator using robustbase package on R. Significance was set at $P \leq 0.05$. State-space modeling was conducted in a Bayesian framework using rstan package on R. All models were assigned default priors. For the parameters used for MCMC sampling, see Methods section and deposited codes.

**Reporting summary**. Further information on research design is available in the Nature Portfolio Reporting Summary linked to this article.

## Data availability
The data that support the findings of this study have been deposited at Open Science Framework (https://doi.org/10.17605/osf.io/tfy9s).

## Code availability
The codes used for the analysis have been deposited at Open Science Framework (https://doi.org/10.17605/osf.io/tfy9s).

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

## Acknowledgements

This study was supported by the Japan Society for the Promotion of Science (grant no. JP16H06324 to T.K. and no. JP21J30006 to K.K.) and the Japan Science and Technology Agency CREST [grant no. JPMJCR17A4 (17941861)] to T.K. We appreciate support from the UTokyo Center for Integrative Science of Human Behavior. We are also grateful for Dr. Ai Kawamori's statistical advice about the state-space modeling.

## Author contributions

K.K., Y.O., A.O., T.T., K.I., and T.K. designed the study. K.K. and Y.O. conducted the experiments. K.K., Y.O., A.O., and T.T. performed the analysis. K.K., Y.O., A.O., and T.K. wrote the manuscript. T.K. supervised the entire process.

## Competing interests

The authors declare no competing interests.
