## [Peer Review File · Communications Biology]

Reviewers' comments:

Reviewer #1 (Remarks to the Author):

This is a well-written paper about a timely topic -- the emergence of shared realities / norms in humans. The authors provide a succinct but adequate literature review, and correctly identify what is currently missing in our understanding of norm formation at the psychophysiological level. The two experiments (behavioral, fMRI) are both well-designed and executed, and the reporting is adequate. Overall, this is an interesting study with clear and novel implications.

I have few comments to consider for possible revision.

1. The justification of the selection of the neural structures is a bit quick.' I fully agree that the TPJ is the key candidate here, yet a few more references could help. I was particularly thinking about a recent paper in PNAS (2018?) on mimicry and economic trust, also revealing a critical role for the TPJ. And there is some work showing interpersonal synchronization in prefrontal structures (e.g., DLPFC) and collective action/social coordination. So perhaps this needs some additional comments?

2. By current standards, sample sizes for both the behavioral and the fMRI studies seem to be on the low side and I was missing any statement on a priori power analyses. Personally I do not need these as hypotheses are clear and results clearly in support as well. However, a comment on this aspect of the methods may be appreciated by the broader readership of Comm Biol.

3. the Behavioral On-line Experiment came in a bit as an afterthought and felt not (really) needed. One possibility is to defer this to the SI and discuss its added insights in the Concluding paragraph with reference to the SI for further detail.

Reviewer #2 (Remarks to the Author):

In today's connected world, new social norms can emerge rapidly within groups. However, little is known about how norms emerge dynamically through social interaction. In the present work, participants estimated the number of dots on the screen, and then saw an estimate provided by a social partner. The authors found that participants' estimates converge to become closer to their partners' after social interaction, and that they converge more closely to reciprocal partners than to non-reciprocal ones. In addition, two regions within the mentalizing network—RTPJ and DMPFC—tracked the participant's similarity to the partner and the estimated number of dots, respectively, and activity in RTPJ correlated with stabilization in participants' estimation weights.

Overall, this is a timely paper, and there were many methodological details in it that I appreciated—the contrast between reciprocal and non-reciprocal agents is subtle and interesting, and I appreciate that the authors identified regions within the mentalizing network using an independent localizer. However, there were several points where I felt that the authors' conclusions were not supported by the results.

Key points:

(1) Lines 44-46: The use of the term "shared reality" creates conceptual and methodological issues that detract from the manuscript's claims. Conceptually, the term is never defined, and does not seem to be doing useful work. If anything, I felt that the use of the term "shared reality" obscured this paper's connections to past work on collective decision-making—this same set of experiments could be

reframed as a project on social conformity, or consensus decision-making. Methodologically, the term implies that social interaction directly changes the participants' percept, as opposed to merely introducing a response bias. As I explain in point 5, below, this interpretation seems too strong, and the data do not tease these alternatives apart.

(2) Lines 86-95: The authors state that a key motivator of the present work is to tease apart "whether bilateral interaction affects only convergence of people's overt behaviors or may also lead to deeper sharing of the covert psychophysical functions that generate their behaviors." There's one possible interpretation of this question that is adequately addressed these experiments, and many stronger ones that are not. I am reasonably convinced by the experiments that, through interaction, participants adopted their partners' response bias, which is deeper than copying than merely copying their partners' answers. However, I am not convinced that this change in response bias reflects a sharing of "deeper covert psychophysical functions"—rather than directly changing percepts, it seems more plausible that participants are anchoring their estimates based on their partners' response bias.

(3) Lines 102-138: The motivation for hypotheses H2-H4 was unclear. What, exactly, drives the stabilization in participants' responses, and why is reciprocity important? Given the authors' appeal to generative models, I was hoping that participants could state these hypotheses more precisely by grounding each of these predictions in their model. For example, one possibility that one could test using generative models is that social information provides an informative prior on people's responses. The stronger the prior, the less further observations would shift participants' posterior beliefs. (See also Bahrami et al., 2010, *Science*, and Campbell-Meiklejohn et al., 2017, *J Neurosci*, for examples of Bayesian generative models that combine perceptual and social information.)

(4) Line 159: The model assumes that participants' responses are systematically biased by some constant weight and linearly related to the true number of dots. For example, a participant with an estimation weight of .90 will systematically underestimate how many dots are on the screen, and the difference between their estimates and the true count should be linearly related to the number of dots on the screen. Is there anything in the behavioral data to suggest that participants' estimates are biased by a stable amount, as the model assumes? Figure 1 only shows the distribution of the estimation weights, which is several steps removed from the raw data. It would be helpful to see, e.g., a curve comparing how many dots are actually on the screen vs. what the participant estimated, for a single representative participant with an estimation weight less than 1. Based on the modeling assumptions, one would expect to see a linear relationship.

(5) Lines 168-180: A key stated goal of this paper is to tease apart whether social information merely affects people's overt responses, or whether it affects their internal perceptions. The results in this section provide convincing evidence that participants' estimates become more similar to one another after interaction, in ways that cannot be account for through mere expertise with the task (as seen by comparison with shuffled pairs). However, I'm not convinced that these analyses actually meet the paper's stated goal. For example, we might expect to see the same results if participants are anchoring their estimates to the estimates of their partners, which would correspond to a response bias rather than a change in the underlying perceptual judgment.

(6) Lines 181-211: In the analysis discussed above, the authors compared the distance in estimation weights between interacting pairs against a shuffled baseline, to account for expertise with the task. I think this analysis—where the authors model stabilization of the estimation weight over time—would benefit from a similar control. In its current form, I am not convinced by the authors' claim that participants' estimation weights stabilized as a result of interaction—one would expect a similar effect from mere expertise with the task, even in the absence of interaction.

(7) Lines 300-302: I'm not convinced that this claim is supported by the fMRI results—I would expect this statement to be backed up, for example, by comparing estimation weights following interaction with a reciprocal or non-reciprocal partner. What information does the fMRI data add that behavioral data alone do not?

(8) Lines 310-311: The authors make a causal claim here about RTPJ involvement that is not supported by the data - I think it would be fair to claim here that there are differences both in participant behavior and in the extent to which RTPJ tracks similarity, based on whether participants are interacting with a reciprocal or non-reciprocal partner. But these data alone are not enough to assign a causal role for RTPJ in guiding the behavior.

Minor points:

(1) General: The paper refers to "Sherif-like" and "Asch-like" agents throughout. The introduction does a good job of motivating the choice of these conditions based on Sherif and Asch's work, but it might be easier to read the methods if these labels were replaced with something more descriptive (e.g., "reciprocal" vs "non-reciprocal" partners).

(2) Line 68, and others: The paper refers to Asch's line estimation task as "a perceptual task without an 'ought' component." What does "'ought' component" mean? It might be clearer to say that these tasks have a verifiable, correct answer - one can check whether the screen had 23 dots or not.

(3) Line 328-330: If participants' estimation weights increased in the non-reciprocal condition, and the non-reciprocal agent is an overestimator, then wouldn't that mean that participants' responses became more similar to those of the non-reciprocal agent? That seems like the opposite pattern of what the authors found in preceding experiments—what do you make of that difference?

(4) Lines 362-363: This is an important detail of these experiments—I wish this were pointed out earlier in the paper, when the task is first introduced.

Reviewer #3 (Remarks to the Author):

Overall, the research reported in this manuscript addresses an important current topic relevant across different fields (cognitive, social, neuroscience), that is, the processes underlying social influence and resulting experiences of shared reality with a partner. The study is well conceptualized, and the experimental methodology strikes me as valid, precise, and appropriate to examine the research questions. I particularly appreciate the relation between neuroscientific evidence and building block processes of shared reality (inference of the partner's inner state via mentalizing; see Echterhoff, Higgins, & Levine, 2009, PPS). Thus, I feel that the manuscript can make a significant contribution. The manuscript would be further strengthened by addressing the following issues.

- Do the findings in this experimental paradigm actually represent as effects of social interaction? Participants do not interact or communicate with each other; they merely receive information of the partner's decisions. In fact, instructions emphasize independence, and participants are explicitly instructed NOT to communicate. The authors should not exaggerate the role of social interaction. Perhaps, it is more appropriate to refer to effects of social influence.

- In this regard, the precision of the conclusions and the anchoring in important literature would be enhanced by discussing whether the present type of influence is normative or informational. This is a relevant classical distinction, and it is generally assumed that Sherif's paradigm reflects predominantly informational influence, while Asch's findings reflect predominantly normative influence. Is this

distinction applicable to the present paradigm? It seems that the results overall suggest greater informational (Sherif-type) influence.

- Regarding another key construct, the authors need to pinpoint and discuss more precisely the extent to which their type of social influence reflects the achievement of shared reality. To this end, full-blown theoretical accounts of shared reality should be considered and cited, primarily, Echterhoff et al. (2009) and Higgins, Rossignac-Milon & Echterhoff (2021, CDPS). My sense is that the present type of social influence should be characterized more precisely on some of the definitional criteria of shared reality, such as subjective / experienced commonality of inner states; relevance to epistemic and relational motives; inner state referring not merely to basic perception but to motivationally relevant responses such as judgments, beliefs, attitudes.

- With the experimental paradigm, one can examine not only how participants' own estimates are affected by social influence, but also how they relate to objective reality (objective accuracy, i.e. difference to the actual number of dots). This aspect is shortly mentioned in relation possible positive effects on accuracy of relative ordering (line 112, line 447) but the issue of effects on accuracy can also be addressed directly by looking at the data: Is there a general tendency to become more accurate over time? If so, does this general tendency additionally benefit from social influence in the behavioural laboratory experiment?

- In the fMRI experiment and in the online behavioral experiment, social influence probably impairs objective accuracy because the simulated partners were deliberately presented as low-performance estimators (strong underestimators or strong overestimators). Given that only "bad" estimators were considered in those two experiments, how can the authors be sure that the same mechanisms would also apply to cases where two estimators within the normal range influence each other (as in Exp.1). It would also be interesting to compare results from Exp.1 for those pairs who deviate strongly vs. weakly in the beginning of the task.

- Line 50: It seems plausible that generative norms are more robust than mere duplications of external norms, but there is no cogent logic that this is necessarily the case. So this is an empirical question. Thus, the authors should provide evidence and/or references.

- The authors investigate simple perceptual norms but they seem to assume that the same mechanisms would apply also to injunctive or value-based (moral) norms (lines 61 ff, lines 454 ff). If this is the case, this assumption should explicitly stated and justified. Because there is a qualitative difference between mere perception and values, such an assumption would need more justification than in the present text.

- The RTPJ, as an ROI related to mentalizing, is more involved in Sherif-type than in Asch-type partner behavior. But wouldn't mentalizing be necessary in both cases equally, because participants would try to understand the other's behaviour regardless of the specific behavior (or probably even more necessary if the other person does NOT show concession behaviors, as one would plausibly expect)?

- The DMPFC, being an ROI related to mentalizing, only tracks the absolute number of dot estimates. But how would this relate to mentalizing, if it is not related to (social) expectations?

- In the functional connectivity analysis, why is the RTPJ defined as seed region, and not the DMPFC?

- In line 259, the authors describe similarity as the parameter of primary interest with regard to Exp.2. Why then was this parameter not analysed in Exp.1?

- Were there order effects in the fMRI experiment? Strong order effects could bias results, and if they are present, it could be useful to compare Sherif vs. Asch conditions additionally between subjects for only the first session.

- p.30: There are two extreme outliers in the left panels of Fig. 3d and Fig 3e. Would the correlations remain significant even without these two outliers?

- The authors state that participants were paid according to their individual performance. How exactly was the payment calculated from their individual performance?

- Why were eye-movements recorded? The results show that the majority of participants first looked at the partner's estimate rather than on the own estimate, but why is this important? Wouldn't the same general findings be expected if participants first look at their own estimate? And if this is an

important aspect, why didn't the authors directly compare participants (or trials) where participants first looked at the partner's estimate with those where they first looked at their own estimate?
- Readability would profit from reduction of jargon and brief explanations for a more general audience (e.g., what are "ICTs" in the first sentence?) would increase.

Reviewer #4 (Remarks to the Author):

In this paper, the authors examined the development of biases in dot estimations through social interaction. Using a dot-estimation task, they were able to model how these interactions change a variable representing whether people over or underestimate the task. Participants interacted either in an "Asch" pair (where the confederate showed no evidence of being influenced by the participant) and in a "Sherif" pair (where the confederate's responses were influenced and changes along with the participant's). Across experiments, participants' dot estimates were influenced by the partner's response primarily in the "Asch" situations, however the stability of these responses was greater in the "Sheriff" situations. In addition, an fMRI study demonstrated that this effect was related to regions implicated in theory of mind.

Below, I list several comments and concerns:

- 1) In the introduction, the authors suggest that it is unknown whether conformity to groups is restricted to overt behavior or whether it reflects some deeper change. This distinction has been studied under the guide of private acceptance vs. public compliance and has also been studied with fMRI (e.g., Zaki et al. 2011).
- 2) In the experiments, it appears that Sherif type pairs had relatively less influence on participant's responses compared to Ash pairs (i.e., $p=0.053$ in one experiment). Instead, the Sherif pairs led to great stability of the weight variable in the model. I find it difficult to parse this finding and wonder at the implication of it. On the one hand, social influence seems greatest in Asch pairs. On the other hand, a person's weighting of dots is more stable in Sherif pairs. Could this 2nd result but simply an artifact of the experiment. In the Asch pairs, the lack of updating on the part of the partner may serve as noise or misinformation that renders estimates more variable, on the other hand, the partner in the Sherif cases serves as another source of variation and something to help aid in decision making. Would the same finding not hold if participants were simply given feedback on the accuracy of their estimates? Would their weights not similarly "stabilize"?
- 3) I'm not entirely sure what novelty the relatively simple linear model brings to the table. We know that conformity and social influence occur. We know that people change their estimates to be in line with the group. Showing this either as changes in their mean response, or as changes in a variable that's related to the mean response (though as a function of the number of dots) doesn't seem to mean much for theory. Put differently, given any reasonable model, would not the weight variable necessarily have to change if we know the mean estimates will change based on the history of research on conformity?

Rebuttal Letter

The new manuscript is composed of a 149-word abstract, 5231-word main text, 88 references, 4 figures, Supplementary Material, 2 supplementary tables, and 12 supplementary figures.

We summarize how we have responded to each comment of the reviewers. The line numbers in our responses below are from the manuscript with **changes highlighted in yellow** (because the Word track changes causes line numbers to skip between pages). Please see the following file: COMMSBIO_Article file_resubmitted (changes highlighted)_09082022.pdf.

Reviewer 1

Comment: (1) The justification of the selection of the neural structures is a bit quick. I fully agree that the TPJ is the key candidate here, yet a few more references could help. I was particularly thinking about a recent paper in PNAS (2018?) on mimicry and economic trust, also revealing a critical role for the TPJ. And there is some work showing interpersonal synchronization in prefrontal structures (e.g., DLPFC) and collective action/social coordination. So perhaps this needs some additional comments?

Thank you for the reference. We have cited the paper by Prochazkova and colleagues (Prochazkova, E. *et al.* Pupil mimicry promotes trust through the theory-of-mind network. *Proc. Natl. Acad. Sci. U. S. A.* **115**, E7265–E7274, 2018) and discussed the critical role of the TPJ with more references. Please see the first paragraph of p. 6 (lines 136–148). Although this study focused on the regions (TPJ, DMPFC) associated with the mentalizing network, we agree with you that other regions such as the DLPFC (e.g., Yang et al., 2020, *Nat. Neurosci.*) may be related to collective-action coordination through interpersonal synchronization. Although the DLPFC did not track trial-by-trial changes in estimation similarity with the Sherif-type partner during interaction (see the results of the whole-brain analysis in Supplementary Fig. 8) as did the RTPJ and DMPFC, we believe that this is an important question to be addressed in a future study.

Comment: (2) By current standards, sample sizes for both the behavioral and the fMRI studies seem to be on the low side and I was missing any statement on a prior power analyses. Personally I do not need these as hypotheses are clear and results clearly in support as well. However, a comment on this aspect of the methods may be appreciated by the broader readership of Comm Biol.

Thank you for the important comment. We have added the following explanation about the

sample sizes and the prior power analysis to the Methods section of the main text.

The behavioral experiments

Lines 485–486; 530–532: The sample size was determined by a power analysis on findings from a previous study employing a similar experimental paradigm⁸⁷ [Murata, A., Nishida, H., Watanabe, K. & Kameda, T. Convergence of physiological responses to pain during face-to-face interaction. *Sci. Rep.* **10**, 450 (2020)].

The fMRI experiment

Lines 504–507: Prior to the experiment, we conducted a power analysis assuming a two-tailed paired *t* test with $\alpha = 0.05$, $\beta = 0.8$, and Cohen's $d = 0.6$, using G*power⁸⁸ (Universität Düsseldorf: G*Power. <http://www.psychologie.hhu.de/arbeitsgruppen/allgemeine-psychologie-und-arbeitspsychologie/gpower.html>). We found that 24 or more participants were needed.

Comment: (3) The Behavioral On-line Experiment came in a bit as an afterthought and felt not (really) needed. One possibility is to defer this to the SI and discuss its added insights in the Concluding paragraph with reference to the SI for further detail.

Following your suggestion, we thought about moving the behavioral online experiment to the SI. However, after revising the other parts, we felt that it might be better suited to the original position and have decided to keep it in the main text.

Reviewer 2

Key points

Comment: (1) Lines 44-46: The use of the term “shared reality” creates conceptual and methodological issues that detract from the manuscript’s claims. Conceptually, the term is never defined, and does not seem to be doing useful work. If anything, I felt that the use of the term “shared reality” obscured this paper’s connections to past work on collective decision-making—this same set of experiments could be reframed as a project on social conformity, or consensus decision-making. Methodologically, the term implies that social interaction directly changes the participants’ percept, as opposed to merely introducing a response bias. As I explain in point 5, below, this interpretation seems too strong, and the data do not tease these alternatives apart.

Thank you for raising these important points. First, about the term “shared reality,” we are sorry that our original explanation of this concept was insufficient, but we think that this concept is useful to understand the emergence of social norms as an example of the dynamic formation of shared realities characterizing the digital era. We argue that socially shared realities affect

what people see/attend to in a focal situation, how and to whom what they have seen is communicated, and when and how they edit their own specific behaviors. Note that this concept covers not only converged perception about the common reference target but also convergence in how to express it in communication (e.g., response bias, selection of terms). According to your and the Editor's suggestions, we have clarified how this concept relates to collective decision making. Please see the first paragraph of the Introduction section (p. 2, lines 44–59). In response to Reviewer 3's comment (3), we have also explained how our experimental results are related to this concept in some detail. Please see the first paragraph of p. 18 (lines 436–456) in the Discussion section.

Second, we agree that our data cannot tease out the distinction between changes in percept and changes in response bias. You are right that our original claim that social interaction directly changes the participants' percept was too strong in this sense. As we explain below (see our responses to your comments 2 & 5), we have used “perceptual responses” instead of “perceptual experiences” throughout the revised manuscript.

Comment: (2) Lines 86-95: The authors state that a key motivator of the present work is to tease apart “whether bilateral interaction affects only convergence of people's overt behaviors or may also lead to deeper sharing of the covert psychophysical functions that generate their behaviors.” There's one possible interpretation of this question that is adequately addressed these experiments, and many stronger ones that are not. I am reasonably convinced by the experiments that, through interaction, participants adopted their partners' response bias, which is deeper than copying than merely copying their partners' answers. However, I am not convinced that this change in response bias reflects a sharing of “deeper covert psychophysical functions”—rather than directly changing percepts, it seems more plausible that participants are anchoring their estimates based on their partners' response bias.

We understand your point that people's decision making can be divided into a cognitive component (e.g., d-prime) and a response component (e.g., decision criterion or response bias) such as modeled by the signal detection theory (Sorkin et al., 2001, *Psych. Rev.*). Although we completely agree that your interpretation is possible, this study was not perfectly designed to address that interpretation; because the experiments did not require participants to make a binary decision, signal detection analysis is not applicable. Yet, your point has led us to reconsider that our interpretation that perception itself changed through interaction may have been too strong. We have thus corrected our previous expressions, stating that social interaction affected participants' “perceptual responses” (e.g., response biases including how to express one's percept to others) rather than “perceptual experiences” per se, throughout the text. Please see lines 34, 86–94, 167, 331, and 373–377 of the revised manuscript.

Comment: (3) Lines 102-138: The motivation for hypotheses H2-H4 was unclear. What,

exactly, drives the stabilization in participants' responses, and why is reciprocity important? Given the authors' appeal to generative models, I was hoping that participants could state these hypotheses more precisely by grounding each of these predictions in their model. For example, one possibility that one could test using generative models is that social information provides an informative prior on people's responses. The stronger the prior, the less further observations would shift participants' posterior beliefs. (See also Bahrami et al., 2010, *Science*, and Campbell-Meiklejohn et al., 2017, *J Neurosci*, for examples of Bayesian generative models that combine perceptual and social information.)

To better motivate our hypotheses H2–H4, we have revised the text substantially. For the possible role of reciprocity in stabilizing participants' responses (H2 and H3), we have discussed previous studies (e.g., Mahmoodi et al., 2018, *Nat. Commun.*) that investigated effects of reciprocity on matching of various social behaviors at the overt level. Please see the second and third paragraphs of p. 5 (lines 119–132). For the activity of the mentalizing network (H4), we have discussed previous studies that showed involvement of these regions to infer the others' agency and mental states as well as to make decisions on behalf of others and argued that the reciprocal agent (“Sherif-type”) would be more likely to be referred to mentally during interaction. Please see the first paragraph of p. 6 (lines 136–148). We appreciate your point about the Bayesian generative models that combined perceptual and social information (e.g., Bahrami et al., 2010, *Science*; Campbell-Meiklejohn et al., 2017, *J. Neurosci.*). However, because we did not employ the before–after design (i.e., first response → social information → second/revised response) in each trial, this analysis is not applicable to our data set. As we have tried an analysis similar to your suggestion elsewhere (Jayles et al., 2017, *Proc. Natl. Acad. Sci. U. S. A.*), we will incorporate your suggestion in our future studies.

Comment: (4) Line 159: The model assumes that participants' responses are systematically biased by some constant weight and linearly related to the true number of dots. For example, a participant with an estimation weight of .90 will systematically underestimate how many dots are on the screen, and the difference between their estimates and the true count should be linearly related to the number of dots on the screen. Is there anything in the behavioral data to suggest that participants' estimates are biased by a stable amount, as the model assumes? Figure 1 only shows the distribution of the estimation weights, which is several steps removed from the raw data. It would be helpful to see, e.g., a curve comparing how many dots are actually on the screen vs. what the participant estimated, for a single representative participant with an estimation weight less than 1. Based on the modeling assumptions, one would expect to see a linear relationship.

Thank you for the useful suggestion. In Supplementary Fig. 1, for each participant in the laboratory behavioral experiment, we have graphed the linear relationship between the number of dots and estimation, as suggested by our linear model [Eq. (1)]. In Supplementary Table 1,

we have also shown a comparison between the linear model and the non-linear (log-linear) model [Eq. (S1)] in terms of the Akaike Information Criterion. As the linear model outperformed the log-linear model, we used the linear model in the analysis. Please see the third paragraph of p. 4 of the SI (“Linear vs. log-linear models of dot estimation”), Supplementary Fig. 1 (p. 19) and Supplementary Table 1 (p. 32).

Comment: (5) Lines 168-180: A key stated goal of this paper is to tease apart whether social information merely affects people’s overt responses, or whether it affects their internal perceptions. The results in this section provide convincing evidence that participants’ estimates become more similar to one another after interaction, in ways that cannot be account for through mere expertise with the task (as seen by comparison with shuffled pairs). However, I’m not convinced that these analyses actually meet the paper’s stated goal. For example, we might expect to see the same results if participants are anchoring their estimates to the estimates of their partners, which would correspond to a response bias rather than a change in the underlying perceptual judgment.

Thank you for raising this important point, which we think echoes your comment (2). We have made it clear in the revised manuscript that Eq. (1) denotes the generative function for behavioral response, $Est_i(t)$, which can reflect both percept and response bias. That is, the estimation weight, w_i , reflects both the cognitive component (e.g., d-prime) and response bias (e.g., decision criterion) as modeled by signal detection theory (Sorkin et al., 2001, *Psych. Rev.*). Please see the second paragraph of p. 7 (lines 166–172). Because our focus was on testing whether such generative functions for behavioral responses might converge through interaction, we did not design the study to address the separation between the two possible components. Given this limitation, we have corrected the overstatement of the previous version, instead stating that social interaction affected participants’ “perceptual responses” rather than “perceptual experiences” throughout the text. Please see lines 34, 86–94, 167, 331, and 373–377 of the revised manuscript.

Comment: (6) Lines 181-211: In the analysis discussed above, the authors compared the distance in estimation weights between interacting pairs against a shuffled baseline, to account for expertise with the task. I think this analysis—where the authors model stabilization of the estimation weight over time—would benefit from a similar control. In its current form, I am not convinced by the authors’ claim that participants’ estimation weights stabilized as a result of interaction—one would expect a similar effect from mere expertise with the task, even in the absence of interaction.

Because the distance in estimation weights between paired participants is a pair-level variable, we conducted a contrast between real pairs and shuffled pairs to see if interaction yielded the convergence (Fig. 1c). In contrast, stabilization of the estimate is an individual-level variable.

Thus, we conducted a contrast between the pair condition and the individual condition to see if interaction increased the stability for each individual (Fig. 1e).

Comment: (7) Lines 300-302: I'm not convinced that this claim is supported by the fMRI results—I would expect this statement to be backed up, for example, by comparing estimation weights following interaction with a reciprocal or non-reciprocal partner. What information does the fMRI data add that behavioral data alone do not?

In accordance with the comment, we redrafted the paragraph to explain neuro-behavioral relations. The fMRI data suggested that participants whose mentalizing network was more involved in referring to the Sherif-type partner showed more stabilized psychological function. Please see lines 323-332 on p. 13.

Comment: (8) Lines 310-311: The authors make a causal claim here about RTPJ involvement that is not supported by the data - I think it would be fair to claim here that there are differences both in participant behavior and in the extent to which RTPJ tracks similarity, based on whether participants are interacting with a reciprocal or non-reciprocal partner. But these data alone are not enough to assign a causal role for RTPJ in guiding the behavior.

In accordance with the comment, we removed the causal claim “This means that reciprocal concession was the key for spontaneous perspective-taking to occur during interaction.”

Minor points

Comment: (1) General: The paper refers to “Sherif-like” and “Asch-like” agents throughout. The introduction does a good job of motivating the choice of these conditions based on Sherif and Asch’s work, but it might be easier to read the methods if these labels were replaced with something more descriptive (e.g., “reciprocal” vs “non-reciprocal” partners).

Thank you. As we feel that these labels convey vivid contrasting images about the two types of partners, we have decided to retain them.

Comment: (2) Line 68, and others: The paper refers to Asch’s line estimation task as “a perceptual task without an ‘ought’ component.” What does “‘ought’ component” mean? It might be clearer to say that these tasks have a verifiable, correct answer - one can check whether the screen had 23 dots or not.

We have corrected the text as suggested. Please see line 65 on p. 3.

Comment: (3) Line 328-330: If participants' estimation weights increased in the non-reciprocal condition, and the non-reciprocal agent is an overestimator, then wouldn't that mean that participants' responses became more similar to those of the non-reciprocal agent? That

seems like the opposite pattern of what the authors found in preceding experiments—what do you make of that difference?

We are sorry but we do not fully understand your point. As you wrote, participants' responses actually became more similar to those of the non-reciprocal (Asch-type) overestimator after interaction (Supplementary Fig. 12), which we do not think to be contradictory to the results with the Asch-type underestimator (Fig. 2b). In contrast, when interacting with the reciprocal (Sherif-type) overestimator, participants retained their underestimation (Supplementary Fig. 12). This may seem contradictory, but this result accrued from the built-in algorithm of the Sherif-type agent that approached participants' estimates rapidly during early phases of interaction (see Supplementary Methods and Eq. (S8) for the algorithm). Please see lines 353–356 on p. 14.

Comment: (4) Lines 362-363: This is an important detail of these experiments—I wish this were pointed out earlier in the paper, when the task is first introduced.

We have provided this information in lines 163–164 on p. 7.

Reviewer 3

Comment: (1) Do the findings in this experimental paradigm actually represent as effects of social interaction? Participants do not interact or communicate with each other; they merely receive information of the partner's decisions. In fact, instructions emphasize independence, and participants are explicitly instructed NOT to communicate. The authors should not exaggerate the role of social interaction. Perhaps, it is more appropriate to refer to effects of social influence.

Thank you for the useful suggestion. We agree with you that our setup allowed only limited exchange of information about the partner's decisions. However, we believe that this situation still qualifies as social interaction, because paired participants in the laboratory behavioral experiment behaved in a reciprocal manner. According to the *APA Dictionary of Psychology* (<https://dictionary.apa.org/social-interaction>), social interaction means “any process that involves reciprocal stimulation or response between two or more individuals.” An experimental setup similar to ours is also common in the collective decision-making literature [see Kameda, T., Toyokawa, W. & Tindale, R. S. Information aggregation and collective intelligence beyond the wisdom of crowds. *Nat. Rev. Psychol.* **1**, 343-357 (2022)]. So, although we understand your concern, we have decided to retain this term.

Comment: (2) In this regard, the precision of the conclusions and the anchoring in important literature would be enhanced by discussing whether the present type of influence is normative or informational. This is a relevant classical distinction, and it is generally assumed that Sherif's

paradigm reflects predominantly informational influence, while Asch's findings reflect predominantly normative influence. Is this distinction applicable to the present paradigm? It seems that the results overall suggest greater informational (Sherif-type) influence.

We agree with you that the type of social influence in our study was essentially informational. Following your suggestions, we have clarified our position in relation to the social influence literature. We have added the following paragraph in the revised manuscript (lines 87–94 on p. 4):

Whereas the Asch and Sherif paradigms can highlight the difference between public compliance and private acceptance and/or between normative and informational social influence (e.g., whether the behavioral convergence persists even after interaction)^{25,32}, here, we aim to address a finer distinction—whether the interaction affects only the convergence of people's overt behaviors or also leads to sharing of the covert psychophysical functions that generate their behaviors. This distinction is critical, because only the latter shared generative model enables endogenous agreement on new targets beyond the initial learning set. We believe that such a generative nature³³ is a fundamental characteristic of social norms.

Comment: (3) Regarding another key construct, the authors need to pinpoint and discuss more precisely the extent to which their type of social influence reflects the achievement of shared reality. To this end, full-blown theoretical accounts of shared reality should be considered and cited, primarily, Echterhoff et al. (2009) and Higgins, Rossignac-Milon & Echterhoff (2021, CDPS). My sense is that the present type of social influence should be characterized more precisely on some of the definitional criteria of shared reality, such as subjective / experienced commonality of inner states; relevance to epistemic and relational motives; inner state referring not merely to basic perception but to motivationally relevant responses such as judgments, beliefs, attitudes.

Thank you for the important suggestion. In the revised manuscript, we have discussed how our findings relate to the notions of shared reality by referring to Echterhoff, Higgins, and Levine (2009, *Perspect. Psychol. Sci.*) and Higgins, Rossignac-Milon and Echterhoff (2021, *Curr. Dir. Psychol. Sci.*). Please see lines 436–456 on p. 18 and Supplementary Fig. 11.

Comment: (4) With the experimental paradigm, one can examine not only how participants' own estimates are affected by social influence, but also how they relate to objective reality (objective accuracy, i.e. difference to the actual number of dots). This aspect is shortly mentioned in relation possible positive effects on accuracy of relative ordering (line 112, line 447) but the issue of effects on accuracy can also be addressed directly by looking at the data: Is there a general tendency to become more accurate over time? If so, does this general tendency additionally benefit from social influence in the behavioural laboratory experiment?

To examine whether participants' objective accuracy improved over time, we performed a 2×2 mixed ANOVA on the average absolute error between the number of dots and dot estimation (between-participant factor: pair vs. individual condition; within-participant factor: Phase 1 vs. 3). None of the main and interaction effects was significant [condition: $F(1, 61) = 0.059$, $P = .809$; phase: $F(1, 61) = 2.384$, $P = .128$; Condition \times Phase: $F(1, 61) = 0.055$, $P = .815$].

Comment: (5) In the fMRI experiment and in the online behavioral experiment, social influence probably impairs objective accuracy because the simulated partners were deliberately presented as low-performance estimators (strong underestimators or strong overestimators). Given that only “bad” estimators were considered in those two experiments, how can the authors be sure that the same mechanisms would also apply to cases where two estimators within the normal range influence each other (as in Exp.1). It would also be interesting to compare results from Exp.1 for those pairs who deviate strongly vs. weakly in the beginning of the task.

To address your point, we computed the correlation coefficient between “how far the partner had deviated from perfect estimation, that is, the absolute error from $w = 1$ in Phase 1” and “how much the error in the participant’s own estimation increased from Phase 1 to 3.” Pearson’s r was 0.42 for the 21 actual pairs in Exp. 1. This r was significantly higher than those from 10,000 sets of shuffled pairs — the null distribution as a control ($P = .008$).

Comment: (6) Line 50: It seems plausible that generative norms are more robust than mere duplications of external norms, but there is no cogent logic that this is necessarily the case. So this is an empirical question. Thus, the authors should provide evidence and/or references.

We have provided references that relate to this argument. Please see lines 53–55 on p. 2.

Comment: (7) The authors investigate simple perceptual norms but they seem to assume that the same mechanisms would apply also to injunctive or value-based (moral) norms (lines 61 ff, lines 454 ff). If this is the case, this assumption should explicitly stated and justified. Because there is a qualitative difference between mere perception and values, such an assumption would need more justification than in the present text.

We agree with you that the previous expression was too strong. We have revised the text accordingly. Please see lines 473–476 on p. 19.

Comment: (8) The RTPJ, as an ROI related to mentalizing, is more involved in Sherif-type than in Asch-type partner behavior. But wouldn’t mentalizing be necessary in both cases equally, because participants would try to understand the other’s behaviour regardless of the specific behavior (or probably even more necessary if the other person does NOT show concession behaviors, as one would plausibly expect)?

We agree with you that mentalizing might have been invoked when participants wondered why the Asch-type partner did not respond to (reciprocate) their concession. But, beyond such overall mentalizing, our time-series analysis revealed that the RTPJ activity tracked trial-by-trial changes in estimation similarity during interaction with the reciprocal Sherif-type, but not with the one-sided Asch-type partner (Fig. 3b). In other words, people appear to have shown much finer and nuanced mentalizing with the Sherif-type than with the Asch-type partner, in response to reciprocal changes in estimation similarity on a trial-by-trial basis. Furthermore, such finer mentalizing process also appear to have contributed to the stabilization of participants' own covert psychophysical functions afterward. Please see lines 399–404 on p. 16.

Comment: (9) The DMPFC, being an ROI related to mentalizing, only tracks the absolute number of dot estimates. But how would this relate to mentalizing, if it is not related to (social) expectations?

We have added the following paragraph to the Discussion section. Please see lines 408–419 on pp. 16–17.

It might be the case that expressing larger estimations in the presence of a partner with a strong underestimation bias was a socially nuanced behavior, which seems to concur with the previous finding that the DMPFC modulates self-related evaluation⁶⁶ embedded in social contexts^{67–69}. Although the RTPJ and the DMPFC are the core areas of the mentalizing network, the functional dissociation of these areas has been proposed⁷⁰: Whereas the RTPJ is thought to facilitate perspective taking, the DMPFC is thought to integrate social information into subjective evaluation^{71–73}. Our results appear to be in line with this view. Last, the functional connectivity between the RTPJ and the DMPFC modulated the stabilization of participants' covert psychophysical functions after interacting with the reciprocal Sherif-type but not with the unresponsive Asch-type partner (H4: Fig. 3e). These regions are known to be recruited even when participants are not explicitly instructed to infer another's mental state^{58,74–76}.

Comment: (10) In the functional connectivity analysis, why is the RTPJ defined as seed region, and not the DMPFC?

As described in the response to Comment (9), there is functional dissociation between the RTPJ and the DMPFC. In terms of cognitive processing as a whole, the DMPFC is thought to integrate social information into subjective evaluation (i.e., behavioral output), and it is the RTPJ that is thought to facilitate perspective taking. Therefore, we used the RTPJ as the seed region in the functional connectivity analysis. Please see lines 302–314 on p. 12–13 and lines 411–415 on p. 17.

Comment: (11) In line 259, the authors describe similarity as the parameter of primary interest with regard to Exp.2. Why then was this parameter not analysed in Exp.1?

As suggested, we analyzed the data from Exp. 1 (laboratory behavioral experiment) using the same method as in Exp. 2 (fMRI experiment). Using the same time-series model as in Exp. 2, we calculated coefficients for similarity ($Coef_{Sim}$) for each participant in the 21 pairs. For each pair, we had classified the two participants into two types based on their pre-interaction response (Phase 1): one with a higher estimation weight (w_i) and the other with a lower w_i . The figure below shows the mean $Coef_{Sim}$ for the higher and lower participant in each pair. As can be seen, those with higher w_i had a negative $Coef_{Sim}$, indicating that they decreased their w_i in response to the preceding similarity. In contrast, those with lower w_i had a positive $Coef_{Sim}$, indicating that they increased their w_i in response to the preceding similarity. The difference between the two types was significant ($t = -3.38, P = .002$). These patterns indicated that participants exhibited a reciprocal response pattern during interaction.

Comment: (12) Were there order effects in the fMRI experiment? Strong order effects could bias results, and if they are present, it could be useful to compare Sherif vs. Asch conditions additionally between subjects for only the first session.

There was no order effect depending on whether the Sherif-type or the Asch-type partner came first. We have also compared the Sherif and Asch conditions using the first encounter only. Although this reduced the sample size by half, the correlations between the brain activity and the stabilization of an individual's covert psychophysical function after interacting were all significant with the Sherif-type partner and not significant with the Asch-type partner.

RTPJ beta and post-interaction stability (corresponding to Fig. 3d)

With the first Sherif-type partner ($n = 15$): $r = -.768, P = .004$

With the first Asch-type partner ($n = 13$): $r = .024, P = .878$

RTPJ-DMPFC functional connectivity beta and post-interaction stability (corresponding to Fig. 3e)

With the first Sherif-type partner ($n = 15$): $r = -.814$, $P < .001$

With the first Asch-type partner ($n = 13$): $r = .410$, $P = .104$

Comment: (13) p.30: There are two extreme outliers in the left panels of Fig. 3d and Fig 3e. Would the correlations remain significant even without these two outliers?

The r s reported for Fig. 3d and e are robust correlations that exclude the influence of outliers. Please see lines 315–322 on p. 13.

Comment: (14) The authors state that participants were paid according to their individual performance. How exactly was the payment calculated from their individual performance?

We have provided this information in the Methods section. Please see lines 497–501 on p. 20.

Comment: (15) Why were eye-movements recorded? The results show that the majority of participants first looked at the partner's estimate rather than on the own estimate, but why is this important? Wouldn't the same general findings be expected if participants first look at their own estimate? And if this is an important aspect, why didn't the authors directly compare participants (or trials) where participants first looked at the partner's estimate with those where they first looked at their own estimate?

We have recorded participants' eye movements to check whether they paid attention to the computer partner's estimates. This was just for a check, and we had no particular hypothesis about which information (their own or the partner's estimation) participants would look at first. Please see the "Attention to partners' estimates" section on pp. 17 in the SI.

Comment: (16) Readability would profit from reduction of jargon and brief explanations for a more general audience (e.g., what are "ICTs" in the first sentence?) would increase.

We have tried to improve the readability of the manuscript by removing jargon (e.g., ICT, etc.) from the text.

Reviewer 4

Comment: (1) In the introduction, the authors suggest that it is unknown whether conformity to groups is restricted to overt behavior or whether it reflects some deeper change. This distinction has been studied under the guide of private acceptance vs. public compliance and has also been studied with fMRI (e.g., Zaki et al. 2011).

In Zaki et al. (2011, *Psychol. Sci.*), participants rated the attractiveness of faces and subsequently learned how their peers ostensibly rated each face. Participants were then scanned using fMRI while they rated each face a second time. The second ratings were influenced by "social norms" (i.e., the peers' average ratings, which were fixed and exogenous for

participants): Participants changed their ratings to conform to those of their peers. This social influence was accompanied by modulated engagement of two brain regions associated with coding subjective value—the nucleus accumbens and the orbitofrontal cortex—a finding suggesting that exposure to social norms affected participants’ neural representations of value assigned to stimuli.

We agree that these findings are important, illustrating the utility of neuroimaging to demonstrate the private acceptance of social norms. However, as seen in the above summary, Zaki et al.’s (2011) study adopted the traditional Asch conformity paradigm, where participants were exposed to the exogenous and fixed “social standards” and had no opportunity to create a shared social reality through mutual influence; the social influence there was unilateral. On the other hand, our focus and motivation in this research are not on the issue of “private acceptance vs. public compliance” in the Asch paradigm but on understanding how a socially shared reality (e.g., norms, social standards) might emerge dynamically through interaction; we thus introduced the Sherif setting that allows bilateral interaction among participants, while using the unilateral Asch setting as a benchmark condition.

Please see the Introduction and especially lines 87–94 on p. 4:

Whereas the Asch and Sherif paradigms can highlight the difference between public compliance and private acceptance and/or between normative and informational social influence (e.g., whether the behavioral convergence persists even after interaction)^{25,32}, here, we aim to address a finer distinction—whether the interaction affects only the convergence of people’s overt behaviors or also leads to sharing of the covert psychophysical functions that generate their behaviors. This distinction is critical, because only the latter shared generative model enables endogenous agreement on new targets beyond the initial learning set. We believe that such a generative nature³³ is a fundamental characteristic of social norms.

Comment: (2) In the experiments, it appears that Sherif type pairs had relatively less influence on participant’s responses compared to Ash pairs (i.e., $p=0.053$ in one experiment). Instead, the Sherif pairs led to great stability of the weight variable in the model. I find it difficult to parse this finding and wonder at the implication of it. On the one hand, social influence seems greatest in Asch pairs. On the other hand, a person’s weighting of dots is more stable in Sherif pairs. Could this 2nd result but simply an artifact of the experiment. In the Asch pairs, the lack of updating on the part of the partner may serve as noise or misinformation that renders estimates more variable, on the other hand, the partner in the Sherif cases serves as another source of variation and something to help aid in decision making. Would the same finding not hold if participants were simply given feedback on the accuracy of their estimates? Would their weights not similarly “stabilize”?

We are puzzled by this comment. Compared to the participants who interacted with the Asch-type partner, participants who interacted with the Sherif-type partner adjusted and stabilized

their judgments over time in response to the partner's judgments (not only at the overt behavioral level but also at the covert psychometric level), but why is this observation “simply an artifact of the experiment”? In addition, we note that “the lack of updating on the part of the Asch-type partner [that] may serve as noise or misinformation” has already been controlled for as a nuisance parameter [A_{typ} in Eq. 2] in our time-series analysis.

We are also puzzled by how this criticism on “artifacts” relates to the following concern: “Would the same finding not hold if participants were simply given feedback on the accuracy of their estimates? Would their weights not similarly ‘stabilize’? We think that providing feedback on accuracy will certainly stabilize the participants’ psychometric functions, but we are unsure about how this might relate to the “artifacts” issue.

Comment: (3) I’m not entirely sure what novelty the relatively simple linear model brings to the table. We know that conformity and social influence occur. We know that people change their estimates to be in line with the group. Showing this either as changes in their mean response, or as changes in a variable that’s related to the mean response (though as a function of the number of dots) doesn’t seem to mean much for theory. Put differently, given any reasonable model, would not the weight variable necessarily have to change if we know the mean estimates will change based on the history of research on conformity?

There seems to be a misunderstanding about the theoretical value of introducing psychometric analysis in our research. As we have stated above, our study is not an extension of the Asch paradigm to study the difference between private acceptance and public compliance (Zaki et al., 2011). We have clarified this point in the Introduction and particularly in lines 87–94 on p. 4.

Thanks the reviewers’ thoughtful comments, we believe that the new manuscript is much improved. We hope that you will agree.

Thank you for your consideration,

Tatsuya Kameda, Ph.D.

Professor

Department of Social Psychology

The University of Tokyo

7-3-1 Hongo, Bunkyo-ku, Tokyo, 113-0033 Japan

Office phone & fax: +81-3-5841-3868

E-mail: tatsuyakameda@gmail.com, tkameda@l.u-tokyo.ac.jp

REVIEWERS' COMMENTS:

Reviewer #2 (Remarks to the Author):

I would like to thank the authors for their work in revising the manuscript. In the revision, they have (1) clarified their use of the term "shared reality," and clarified how their experimental results relate to this construct, (2) softened their claims about the extent to which participants' responses are driven by changes in perception vs. response biases, and (3) expanded the introduction to clarify the motivations for each specific hypothesis. I believe that these changes have adequately addressed the points in my review.

Reviewer #3 (Remarks to the Author):

As I had said in the first review round, the topic of this paper is an important one straddling different fields (cognitive, social, neuroscience). My sense is that the authors have responded carefully to the issues that were raised. As such, the revision makes a big step towards publishability. I only have a few remaining comments:

- The concept of shared reality and social norm should not be used synonymously. Shared reality is a motivated commonality of judgments, beliefs etc. among individuals or within groups about a current object or state of affairs. People having a shared reality privately believe in the truth or validity of their (shared) inner states. Norms are typically defined as shared group standards of *behavior*. As such, norms regulate how one should respond to a given situation, and they need not be privately internalized. Norm internalization is a late step, following formation and dissemination of a norm. Shared reality involves the experience of a personal connection (e.g., Echterhoff & Higgins, 2017, ERSP). Thus, shared reality is different from mere alignment with social norms or informational social influence, which does not require a personal connection to sources of influence. For instance, with identical informational input from a remote communication partner, a shared reality that has been apparently created with a partner is eliminated when communicators subsequently learn that the person who actually received their message is not the person to whom they intended to send their message (Echterhoff, Kopietz, & Higgins, 2013, Exp. 2 & 3, Social Cognition).

- Thus, in line 45, for instance, it should say "shared realities AND norms". This may sound picky but the two concepts do differ.

- line 38: Isn't interaction always, at least, "bilateral"? The attribute could be deleted.

- line 120: The authors say: "We conjectured that reciprocity would be a key to facilitating covert-level convergence, as distinct from overt-level convergence, in behavior." This seems to imply that reciprocity does not facilitate overt convergence. But this does not necessarily have to be the case. Reciprocity in dynamic interactions such as joint physical activities or sports seems key to creating overt behavioral convergence.

- line 123: it should say "self-disclosure" rather than "self-exposure."

Reviewer #4 (Remarks to the Author):

I have read the authors' responses to my own and the other reviewers' comments and believe they have done a commendable job. I have no further concerns or comments.

November 18, 2022 (Manuscript Number: COMMSBIO-22-0393A)

Dear Dr. Inglis,

Thank you for your and your reviewers' thoughtful comments on our manuscript "Behavioral and neuro-cognitive bases for emergence of norms as socially shared realities via dynamic interaction" for publication in *Communications Biology*. We are pleased that our manuscript will be publishable upon final revision in response to Reviewer 3's comments. Below we briefly summarize how we have responded to each comment from Reviewer 3.

Comment: The concept of shared reality and social norm should not be used synonymously. Shared reality is a motivated commonality of judgments, beliefs etc. among individuals or within groups about a current object or state of affairs. People having a shared reality privately believe in the truth or validity of their (shared) inner states. Norms are typically defined as shared group standards of *behavior*. As such, norms regulate how one should respond to a given situation, and they need not be privately internalized. Norm internalization is a late step, following formation and dissemination of a norm. Shared reality involves the experience of a personal connection (e.g., Echterhoff & Higgins, 2017, ERSP). Thus, shared reality is different from mere alignment with social norms or informational social influence, which does not require a personal connection to sources of influence. For instance, with identical informational input from a remote communication partner, a shared reality that has been apparently created with a partner is eliminated when communicators subsequently learn that the person who actually received their message is not the person to whom they intended to send their message (Echterhoff, Kopietz, & Higgins, 2013, Exp. 2 & 3, Social Cognition).

Thus, in line 45, for instance, it should say "shared realities AND norms". This may sound picky but the two concepts do differ.

We agree to the reviewer. We have corrected the text as suggested. Please see line 49 in the final manuscript.

Comment: line 38: Isn't interaction always, at least, "bilateral"? The attribute could be deleted.

We have changed "bilateral interaction" to "bilateral influence", and "unilateral interaction" to "unilateral influence" throughout the manuscript. Please see lines 42, 82, and elsewhere.

Comment: line 120: The authors say: "We conjectured that reciprocity would be a key to facilitating covert-level convergence, as distinct from overt-level convergence, in behavior." This seems to imply that reciprocity does not facilitate overt convergence. But this does not necessarily have to be the case. Reciprocity in dynamic interactions such as joint physical

activities or sports seems key to creating overt behavioral convergence.

We have deleted “, as distinct from overt-level convergence, in behavior.” Please see line 124.

Comment: line 123: it should say “self-disclosure” rather than “self-exposure.”

We have changed the text as suggested. Please see line 128.

We have also edited our manuscript to comply with the format requirements of *Communications Biology*. Again, we thank you and your reviewers for the constructive comments.

Sincerely yours,

Tatsuya Kameda, Ph.D.

Professor

Department of Social Psychology

The University of Tokyo

7-3-1 Hongo, Bunkyo-ku, Tokyo, 113-0033 Japan

Office phone & fax: +81-3-5841-3868

E-mail: tatsuyakameda@gmail.com, tkameda@1.u-tokyo.ac.jp